# Early elevated IFNα is a key mediator of HIV pathogenesis
Hélène Le Buanec[1], Valérie Schiavon[1], Marine Merandet[1], Alexandre How-Kit[2], David Bergerat [1],
Céline Fombellida-Lopez[3], Armand Bensussan [1], Jean-David Bouaziz[1,4], Arsène Burny[5,6], Gilles Darcis[3],
Hongshuo Song[7], Mohammad M. Sajadi[6,7,8], Shyamasundaran Kottilil[6,7,8], Robert C. Gallo [6,7,10] & 
Daniel Zagury[9,10]

## Abstract

**Background** A complete understanding of the different steps of HIV replication and an effective drug combination have led to modern antiretroviral regimens that block HIV replication for decades, but these therapies are not curative and must be taken for life. "Elite controllers" (ECs) is a term for the 0.5% of HIV-infected persons requiring no antiretroviral therapy, whose status may point the way toward a functional HIV cure. Defining the mechanisms of this control may be key to understanding how to replicate this functional cure in others.
**Methods** In ECs and untreated non-EC patients, we compared IFNα serum concentration, distribution of immune cell subsets, and frequency of cell markers associated with immune dysfunction. We also investigated the effect of an elevated dose of IFNα on distinct subsets within dendritic cells, natural killer cells, and CD4+ and CD8 + T cells.
**Results** Serum IFNα was undetectable in ECs, but all immune cell subsets from untreated non-EC patients were structurally and functionally impaired. We also show that the altered phenotype and function of these cell subsets in non-EC patients can be recapitulated when cells are stimulated in vitro with high-dose IFNα.
**Conclusions** Elevated IFNα is a key mediator of HIV pathogenesis.

## Plain language summary

Currently, HIV infection is not curable, but infected individuals can manage their condition by taking daily doses of antiretroviral therapy. Some individuals, known as elite controllers (ECs), control their infection without antiretroviral treatment, and studying how their immune system responds to HIV exposure could lead to a potential cure for others. Here, we compare immune cell responses between ECs and untreated non-ECs. We find that IFNα, a small protein with an important role in controlling white blood cell activity, is produced in excess in immune cells from non-ECs compared with ECs during early infection. This insight provides an important clue for the future development of a targeted cure for HIV.

Potent antiretroviral therapy (ART) was chiefly developed through advances in understanding the stages and molecular events involved in HIV replication. However, these advances have not led to a "functional cure" in which no further therapy is needed to suppress virus resurgence and immune decline. One potential solution is the development of longer-lasting drugs and better delivery. An additional approach is to gain a greater understanding of HIV pathogenesis. A small group of HIV-infected patients have an undetectable viral load without therapy and are known as elite controllers (EC) (<0.5%). This patient group was identified by several clinical investigators and highlighted by ref. [1], who suggested that perhaps

EC status could be mimicked to reach a "functional cure"[2]. Individuals with a genotype including $HLA_{B57+}$ ($EC_{B57+}$), which shares the Bw4 serotype with the KIR3DL1 allele, were found to be more prone to EC status[3]; however, not all ECs are $HLA_{B57+}$[4]. Consequently, other mechanisms must contribute to the EC status.

Given our long-term studies on the pathogenic (high levels) effects of IFNα in AIDS[5,6], we questioned whether control of the pathogenic effects of IFNα could be a major immune mechanism in ECs. Indeed, our initial and other reported studies show that elevated IFNα exerts several pathogenic effects. These effects arise first at the innate phase of an immune response by

[1]Université de Paris; INSERM U976, HIPI Unit, Institut de Recherche Saint-Louis, F-75010 Paris, France. [2]Laboratory for Genomics Foundation Jean Dausset-CEPH, Paris, France. [3]Laboratory of Infectious Diseases, GIGA-I3, GIGA-Institute University of Liege, 4000 Liege, Belgium. [4]Dermatology Department, Hôpital Saint-Louis, Assistance Publique-Hôpitaux de Paris (AP-HP), Paris, France. [5]Laboratory of Molecular Biology, Gembloux Agrobiotech, University of Liège, Liège, Belgium. [6]Global Virus Network, Baltimore, MD 21201, USA. [7]Institute of Human Virology, School of Medicine, University of Maryland, Baltimore, MD 21201, USA. [8]University of Maryland School of Medicine, Baltimore, MD 21201, USA. [9]21CBIO, Paris, France. [10]These authors jointly supervised this work: Gallo RC, Zagury D. ✉e-mail: rgallo@ihv.umaryland.edu

inhibiting IL-7–induced T cell proliferation, controlling T cell homeostasis[7]. They then emerge at the initiation phase of the adaptive immune response by inhibiting the proliferation of CD4+T helper cells[8], followed by their effects on T cell differentiation into suppressive IL-10-Tr1 cells[9,10]. Systemic IFNα (type I IFN) and mucosal IFN-lambda (IFNλ) (type III IFN) both exert antiviral activities via IFN-stimulated genes[11], although with different kinetics of gene induction and distinct cell targets expressing specific receptors. IFNAR1 and IFNAR2 are constitutively expressed on virtually all nucleated cells, such as CD4+T cells, whereas at homeostasis, IFNλ receptors are constitutively expressed on epithelial cells and only in a selected pool of immune cells[12]. Consequently, in inflammatory HIV lymphoid tissues, elevated IFNα locally hampers CD4+T helper and CD4+ follicular T helper cell function and thus inhibits the initiation phase of the anti-HIV adaptive immune response, leading to the development of HIV-specific neutralizing antibody responses. Even at a high level, IFNλ does not hamper CD4+T cell–induced initiation of the immune response. Of note, a 2023 report characterized the efficacy of therapeutic administration of IFNλ in patients in the early stages of COVID-19[13].

The conclusion that elevated IFNα is a central mediator of HIV pathogenesis is supported by previous findings and the current results. Genetic studies have identified polymorphisms in IFN type 1 and type 3 pathways as contributing to variable responses to HIV infection[14]. In addition, SIV-infected African green monkeys do not express elevated levels of IFNα and do not develop AIDS[15]. In contrast, SIV-infected rhesus macaques develop AIDS and have high serum IFNα[16]. Results from the EURIS phase 2B placebo-controlled clinical trial, carried out in five hospital centers from 1996–1998 in 240 patients who had not received combined ART, were based on anti-IFNα antibody induction in an active vaccine approach to lower IFNα. All study endpoints were met among vaccine responders, including a lower occurrence of HIV-related events, improved CD4+T cell counts, and reduced viral load correlating with the rise of anti-IFNα antibodies[17]. Finally, we show here that IFNα induces a series of immune cell abnormalities in untreated non-EC patients.

We find that untreated patients with HIV (non-ECs) have abnormally high concentrations of serum IFNα and a high frequency of phenotypic alterations in all immune cell types. These alterations are known effects of IFNα. In contrast, ECs show minimal abnormal immune cell alterations. We further identify potential immune mechanisms enabling each EC to control viral replication through individual immune capacity. Finally, we hypothesize that EC control of IFNα is at least partially circumstantial, relying on infection with only a low dose of HIV. Because IFNα levels in the primary phase of the acute infection correlate with virus titer, we propose that the critical chain of events in uncontrolled virus replication in non-ECs is a vicious cycle of HIV → IFNα → more HIV → elevated IFNα → more HIV, and so on. In an accompanying manuscript, we described the mechanism leading to this ongoing cycle[18].

## Methods
### Human samples
A total of 73 Healthy donor (HD) samples were obtained through Etablissement Français du Sang (EFS, Paris, France). Fifty-one samples were used for serological analysis and 22 for phenotypic analysis. A total of 48 people living with HIV were recruited and subdivided into two groups: ECs (n = 18), from the Natural Viral Suppressors Cohort (Baltimore), and non-ECs (n = 30), from the National Institutes of Health (Bethesda n = 19) and from the Laboratoire de Référence SIDA (Liège n = 11). Patient groups did not significantly differ in terms of age, gender, or disease status. All participants provided informed consent in accordance with protocols approved by the regional ethical research boards and the Declaration of Helsinki (H-29331 (USA) and 2020/418 (Belgium)). Clinical data are given in Supplementary Table 1.

### Sample processing
Peripheral blood and serum were collected into appropriate tubes. Peripheral blood mononuclear cells (PBMCs) were isolated by density gradient centrifugation on Ficoll-Hypaque (Pharmacia, St Quentin en Yvelines, France). PBMCs were stored frozen in liquid nitrogen.

### Cell culture
CD4+T cells were isolated from frozen PBMCs. All CD4+T cells were positively selected with a CD4+T cell isolation kit (Miltenyi Biotec, Germany), yielding CD4+T cell populations at a purity of 96–99%. Purified CD4+T cells were stimulated with 4 μg/mL plate-bound anti-human CD3 (OKT3) monoclonal antibody (mAb; eBioscience, San Diego, CA, USA) and 4 μg/mL soluble anti-human CD28 (CD28.2) mAb (Becton Dickinson, Le Pont De Claix France) in the presence of recombinant human IL-2 (Proleukine, Chiron, Amsterdam, 100 U/mL) and recombinant human interferon alpha-2a (Roferon-A) and IFNλ2 (Biotechne, UK) at the indicated dose[19]. After 5 days of culture, CD38 and CD25 expression and the frequency of 7-AAD+ cells were measured by flow cytometry on stimulated CD4+T cells. Natural killer (NK) cells were isolated from PBMCs. NK cells were negatively selected with the NK cell isolation kit (Miltenyi Biotec, Germany), yielding NK cell populations at a purity of 96–99%. NK cells were stimulated with IL-15 (Miltenyi Biotec, 10 ng/mL), IL-2 (Proleukine, Chiron, Amsterdam 100 U/mL), and recombinant human interferon alpha-2a (Roferon-A) at the indicated dose. After 3 days of culture, expression of CD56, CD95, and NKG2D was measured by flow cytometry (Supplementary Table 2a).

### Antibody panels, staining, and flow cytometry analysis
Immunophenotypic studies were performed on frozen samples, using flow cytometry panels with up to 23 colors. Approximately $1 \times 10^6$ to $5 \times 10^6$ frozen PBMCs were used per patient per stain. Staining was performed as follows, except for the panel using HLA-E or HLA-A*02 pentamers (Proimmune). PBMCs were thawed using RPMI supplemented with 20% FCS per standard protocol. Cells were resuspended in 45 μL FACS buffer (PBS supplemented with 0.5% BSA and 2 mM EDTA). To block Fc receptor binding, 5 μL of FcR Blocking Reagent (Miltenyi Biotec, Paris, France) was added to the cells for 10 minutes at 4 °C, following the manufacturer's instructions. Cells were washed once with FACS buffer before the staining. Staining was performed in several stages, depending on the number of antibodies. The first mAb cocktail for surface staining, prepared with brilliant ultraviolet, brilliant violet, and brilliant blue antibodies in Brilliant Stain Buffer Plus (BD Biosciences, Paris, France), was added to the cells, followed by incubation for 30 min at 4 °C (see Supplementary Table 2 for antibody panel information). Cells were washed once with FACS buffer before the second mAb cocktail was added, which contained all other antibodies in FACS buffer for surface staining and viability staining, followed by further incubation for 30 min at 4 °C. When needed, cells were fixed and permeabilized using the Foxp3 Staining Buffer Set (eBioscience, Paris, France) according to the manufacturer's protocol. Cells were then stained for intracellular targets with the third mAb cocktail in the kit's permeabilization buffer. After incubation, cells were washed with permeabilization buffer and FACS buffer. When using HLA-E or HLA-A*02 pentamers (ProImmune), pre-incubation with a blocking anti-CD94 mAb (clone HP-3D9, 5 μg/mL, BD Biosciences) was performed to completely abrogate the non-specific staining of CD94+/NKG2+ T cells by HLA-E pentamers before the staining with the other antibodies. Before use within this panel, all antibodies were titrated individually to identify the concentration that provided the maximal brightness of the positive cell population and the lowest signal for the negative cell population. Cells were acquired on a Cytek Aurora flow cytometer. Data were analysed using FlowJo software (FlowJo, LLC). Flow cytometry data were identified as the proportion (%) of cells expressing the marker, and protein levels as median fluorescence intensity. The gating strategy used to identify the immune cell subtypes and their respective subsets is depicted in Supplementary Fig. 1. Unsupervised analyses were performed using Cytobank software and R studio software.

### 7-AAD (7-amino-actinomycin D) staining
Apoptosis of stimulated CFD-labeled CD4+T cells was determined using the 7-AAD assay[20]. Briefly, cultured cells were stained with 20 μg/mL nuclear

dye 7-AAD (Sigma-Aldrich) for 30 min at 4 °C. FSC/7-AAD dot plots distinguish living (FSC^high/7-AAD−) from apoptotic (FSChigh/7-AAD+) cells and apoptotic bodies (FSClow/7- AAD+) and debris (FSClow/7-AAD−). Living cells were identified as CD3+7-AAD−FSC+ cells.

## CellTrace violet staining
CD4+T cells and NK cells were stained with 1 μM dye (CellTrace violet; Molecular Probes/Invitrogen) in PBS for 8 min at 37 °C at a concentration of $1 \times 10^6$ cells/mL. The labeling was stopped by washing the cells twice with RPMI-1640 culture medium containing 10% FBS. The cells were then resuspended at the desired concentration and subsequently used for proliferation assays.

## Composite cell phenotypic alteration score
We generated a cumulative phenotypic score for each T cell subset. The frequency of the following markers was used to calculate the score: CD25−, CD26−, HLA-DR+, CD38+, CTLA-4+, CD28−, PD1+, and CD39+. The score is calculated as the sum of the ratio of the expression level of each marker to the average expression level of the corresponding marker in HDs.

## Cytokine quantification
Serum IFNα and IFNλ2 levels were determined using SIMOA cytokine assays (references 100860 and 101419, respectively). Van der Sluis et al. showed that HIV infection of co-cultures of CD4+T cells and pDCs enhances mRNA expression of IFNλ2 and not IFNλ1 or IFNλ3, so we focused only on serum IFNλ2 levels[21]. IL-10 in 4-day cell culture supernatants was determined by Luminex technology (human custom Procarta Plex, Invitrogen).

## Statistics and reproducibility
Comparisons between the two groups were made using the Mann–Whitney *U*-test. Multiple group comparisons were made using the Kruskal–Wallis test with Dunn's multiple comparison testing. Error bars on graphs represent interquartile ranges. Correlations were assessed using the nonparametric Spearman test. Analyses were performed with GraphPad Prism and R. A two-sided $p$ value less than 0.05 was considered statistically significant (ns: not significant; $*p < 0.05$; $**p < 0.01$; $***p < 0.001$; $****p < 0.0001$).

## Reporting summary
Further information on research design is available in the Nature Portfolio Reporting Summary linked to this article.

# Results
As detailed in the methods, in untreated non-ECs, ECs, and HDs, we compared IFNα and IFNλ2 serum concentration, the distribution of immune cell subsets, and the frequency of cell markers associated with immune dysfunction. These endpoints include abnormal expression of checkpoint receptors or molecules controlling immune cell function such as inhibitory checkpoint receptors (PD1, CTLA-4) that lead cells to the exhaustion stage[22]; cell receptor markers of activation/differentiation (CD25/CD38/HLA-DR) associated with chronic immune activation[23]; checkpoint receptors controlling pericellular levels of immunosuppressive adenosine (CD39 ectonucleotidases, CD26 harboring adenosine deaminase)[24]; apoptotic cell receptors (CD95); and soluble suppressive mediators (IL-10). This phenotypic study focuses particularly on cytotoxic NK cells, HLA-1a-(B)–restricted CD8+ cytotoxic T lymphocyte (CTL) cells, and HLA-1b-(E)–restricted CD8+ suppressive cells (CD8+supps). Of interest, CD8+supp cells, which lyse abnormal CD4+T cells expressing HIV peptides in the HLA-E–restricted context, also are present in other chronic inflammatory diseases, including autoimmune diseases[25], cancer[26], and other microbial infections[27].

## IFNα is elevated in HIV-infected non-ECs and modifies the distribution of blood immune cell types
Principal component analysis (PCA) of blood immune cell type distribution among non-ECs, ECs, and HDs shows that non-ECs and HDs cluster apart, whereas ECs overlap with both of the other groups (Fig. 1a). Figure 1b, c

show the distinct proportion of circulating immune cell types in non-ECs, ECs, and HDs. We find that the serum concentration of IFNα but not of IFNλ2 is abnormally increased in non-ECs compared with HDs and the large majority of ECs (Fig. 1d). We also find a positive correlation between IFNλ2 and IFNα serum levels in ECs but not in non-EC patients (Fig. 1eI). Moreover, serum IFNλ2 concentration is negatively associated with CD4+T cell frequency and positively associated with CD8+T cell frequency in non-ECs (Fig. 1eII, eIII).

## Elevated IFNα induces an abnormal frequency and phenotypic alterations of NK cells and other innate cell types in non-ECs but not in ECs
NK cells are a heterogeneous population of immune cells[28], as SPADE analysis (Fig. 2aI) and dot plots (Fig. 2aII) demonstrate. As previously reported[29], we find that non-ECs have fewer early and mature NK cells than do HDs (2.58 vs 4.61% and 48.1 vs 83.2%, respectively), and many more terminal NK cells (17.91 vs 2.84%) (Fig. 2aIII). ECs and HDs have similar proportions of NK cell subsets. In the mature NK subset, the activating markers Helios, natural cytotoxicity receptors, and GrzB/perf are less frequent in non-ECs than in ECs and HDs (Fig. 2bI, bII). Inhibitory killer-cell immunoglobulin-like receptors (iKIRs) bind HIV peptides present on HLA-1b in activated CD4+T cells as well as the inhibitory checkpoint receptors PD1 or CD39, leading cells to an exhaustion-like status[22]. We find that iKIRs are overexpressed in non-ECs compared with ECs and HDs (Fig. 2bIII, IV). Of note, CD38 and HLA-DR are prevalent in non-EC mature NK cells. In addition, IFNα levels correlate positively with the frequency of iKIR+ mature NK cells (Fig. 2cI), terminal NK cells (Fig. 2cII), and iKIR+ terminal NK cells (Fig. 2cIII). This finding is consistent with the concept that IFNα enhances the development of non-functional NK cells. Furthermore, indicative of loss of function among NK cells in non-ECs, we find that the activation marker NKG2D is only weakly expressed on early NK cells (Fig. 2dI) in this group and that the apoptotic marker CD95 is highly expressed on mature NK cells (Fig. 2dII), respectively reducing their cytotoxic activity and increasing their propensity to apoptosis.

To investigate whether elevated IFNα induces these altered NK cell subset distribution and surface phenotypes, we treated in vitro–purified NK cells with increasing doses of IFNα. We find that elevated IFNα inhibits NK cell viability and proliferation capability in a dose-dependent manner (Fig. 2eI, eII). Furthermore, elevated IFNα reduces CD56 expression in CD56^dim/negNK cells (Fig. 2eIII), leading to a decrease in the frequency of mature NK cells and a concomitant increase in the percentage of terminal NK cells (Fig. 2eIV). The addition of increasing amounts of IFNα to NK cell culture induces a dose-dependent decrease in NKG2D expression in early NK cells (Fig. 2eV) and an increase in CD95 expression in mature NK cells (Fig. 2eVI). In summary, elevated IFNα levels can, in large part, induce the NK cell abnormalities observed in non-ECs.

Regarding other innate immune cells, the frequency of CD11c−/CD123+ plasmacytoid dendritic cells (pDCs) is lower in non-ECs than in ECs (13.4 versus 32.5%, respectively) (Supplementary Fig. 2a). This observation was previously made by others[30,31]. γδ T cells show an abnormally high expression of inhibitory checkpoint receptors in non-ECs (Supplementary Fig. 2b). The expression of these checkpoint receptors, including CD38[32] and HLA-DR[32], is triggered directly by IFNα and indirectly by loss of CD26[19].

## Elevated IFNα alters immune cell homeostasis in non-ECs compared with HDs and a majority of ECs
Homeostatic disturbances of T cell subsets after HIV infection are hallmarks of early disease. From our investigation of the maintenance of CCR7+CD4+ and CD8+T cells in the three groups, as anticipated, we observe a lower frequency of both T cell subsets in non-ECs compared with ECs and HDs (Supplementary Fig. 2cI, cII). In addition, we find negative correlations between CD4+ and CD8+ central memory (CM) T cell frequency and serum IFNα levels in HIV patients (ECs + non-ECs) (Supplementary Fig. 2cIII, cIV). This result reflects the pathogenic inhibitory effects of IFNα on IL-7–induced homeostatic T cell proliferation[7] in CD4+ and CD8+CCR7+T cells homing to

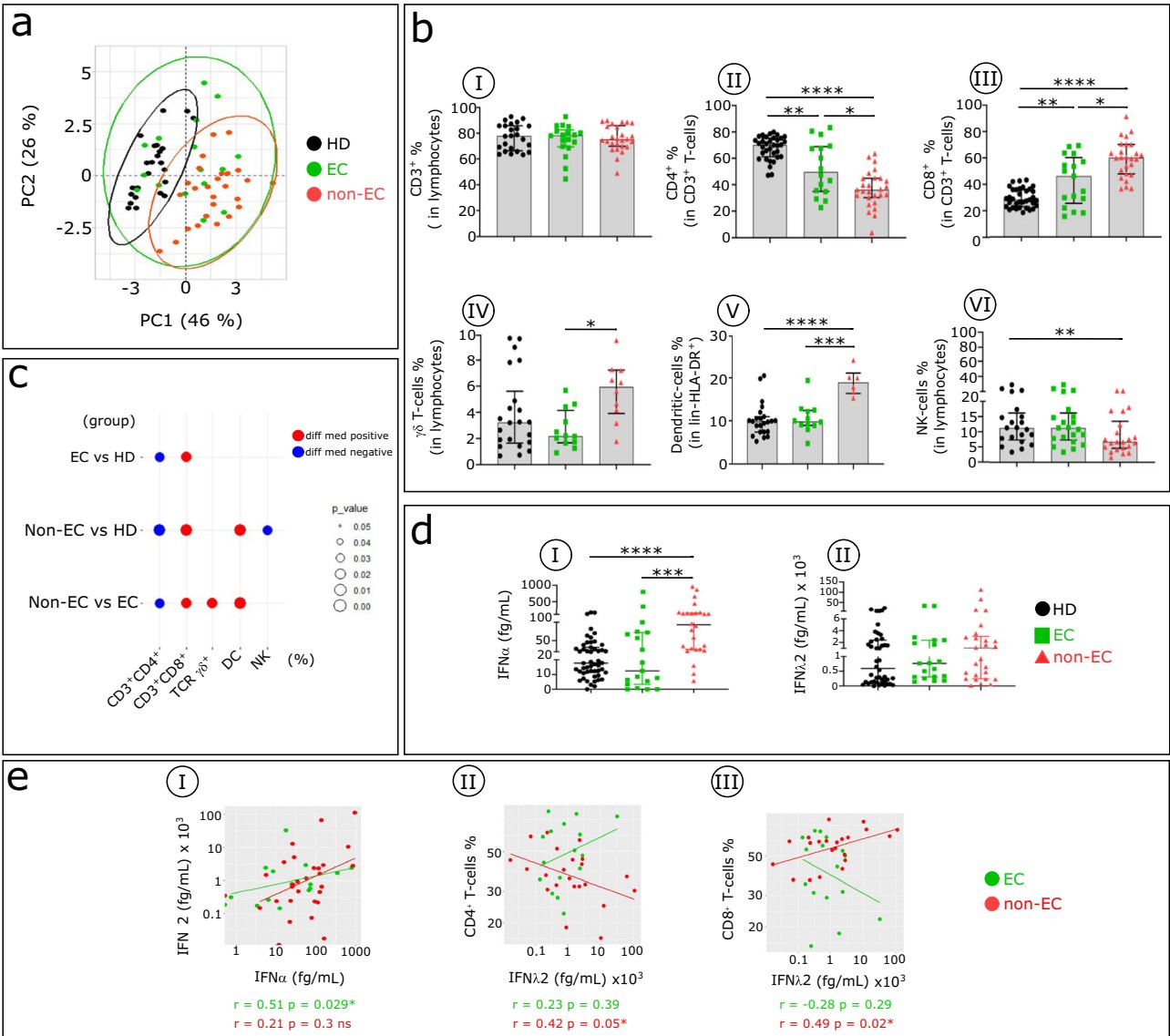

**Fig. 1 | Comparative analysis of major blood immune cell subsets and serum IFNα and IFNλ2 concentrations in non-ECs, ECs, and HDs. a** PCA of data based on the proportion of different immune cell subpopulations (CD4+, CD8+, and TCR γδ T-cells, NKs, and DCs), evaluated by flow cytometry. Immune cell profiling was assessed by flow cytometry, as depicted in Supplementary Fig. 1. The first two principal components (PC1 and PC2) explaining the greatest differences among individuals are represented on a bi-plot. Each point represents one participant, with colors indicating the group: HDs (black), ECs (green), and non-ECs (red). Each group is outlined by an ellipse representing the 95% confidence interval of the sample groupings. **b** Histograms show distributions of indicated immune cell populations between HDs (black), ECs (green), and non-ECs (red). Analysis was done in 24 HDs, 16 ECs, and 26 non-ECs for all populations except for γδ T cells and DC (lin⁻HLA-DR+) (22 HDs, 12 ECs, 10 non-ECs). **c** Balloon plot summarizes the statistically

significant changes in the indicated immune cell populations between ECs and HDs, non-ECs and HDs, and non-ECs and ECs. The size of the circle represents the $p$ value. Red and blue indicate increased or decreased frequencies of immune cell populations. **d** Scatterplots show IFNα and IFNλ2 concentration in serum from HDs ($n = 51$), ECs ($n = 18$), and non-ECs ($n = 26$). IFNα and IFNλ2 levels were detected using SIMOA. **e** Scatterplot of relationships between IFNα and IFNλ2 serum levels (**eI**), CD4+T cells and IFNλ2 (**eII**), and CD8+T cells and IFNλ2 (**eIII**) in ECs ($n = 18$) and non-ECs ($n = 26$). Correlations were evaluated using Spearman's rank correlation test. Multiple group comparisons were assessed using the Kruskal–Wallis test with Dunn's multiple comparison testing. Values are medians and $p$ values (*$p < 0.05$, **$p < 0.01$, ***$p < 0.001$, ****$p < 0.0001$). Error bars on graphs represent interquartile ranges.

lymph nodes[33]. Interestingly, even though the frequency of EC_B57+ CD8+CM cells is similar to that of HDs, the frequency of EC_B57− CD8+CM cells is reduced, albeit to a lesser degree than in non-ECs (Supplementary Fig. 2cV and Table 1). The maintenance of T cell homeostasis observed in EC_B57+ but not in EC_B57− cases (Table 1) may be accounted for by the homology between HLA_B57+ and NK iKIR (specifically the 3DL1 allele) sharing the bw4 serotype. Therefore, iKIR+ NKs do not recognize HLA_B57 peptide epitopes, and NK cell killing is not inhibited, which enables the unaltered EC_B57+ iKIR+ NK cell lysis of infected CD4+T cells, resulting in reduced viral load and a correlated IFNα production level[34]. Consequently, upon TCR stimulation, the percentage of

CM T cells that differentiate into CD4+T helper and cytotoxic CD8+T cells declines in non-ECs but not in EC_B57+ cases.

In addition to the direct killing of some CD4+T cells by HIV, early HIV regulatory proteins Nef and Tat are released during the acute phase of infection. Nef reduces cellular HLA-1a expression[35], and Tat enhances IFNα production by macrophages[5], thus contributing along with high HIV levels to elevated IFNα. This effect, in turn, leads to phenotypic and functional alterations in NK cells. The outcome is a lack of early lysis of infected CD4+T cells in most HIV-infected patients, as we show here. These immune cell alterations are spared in ECs.

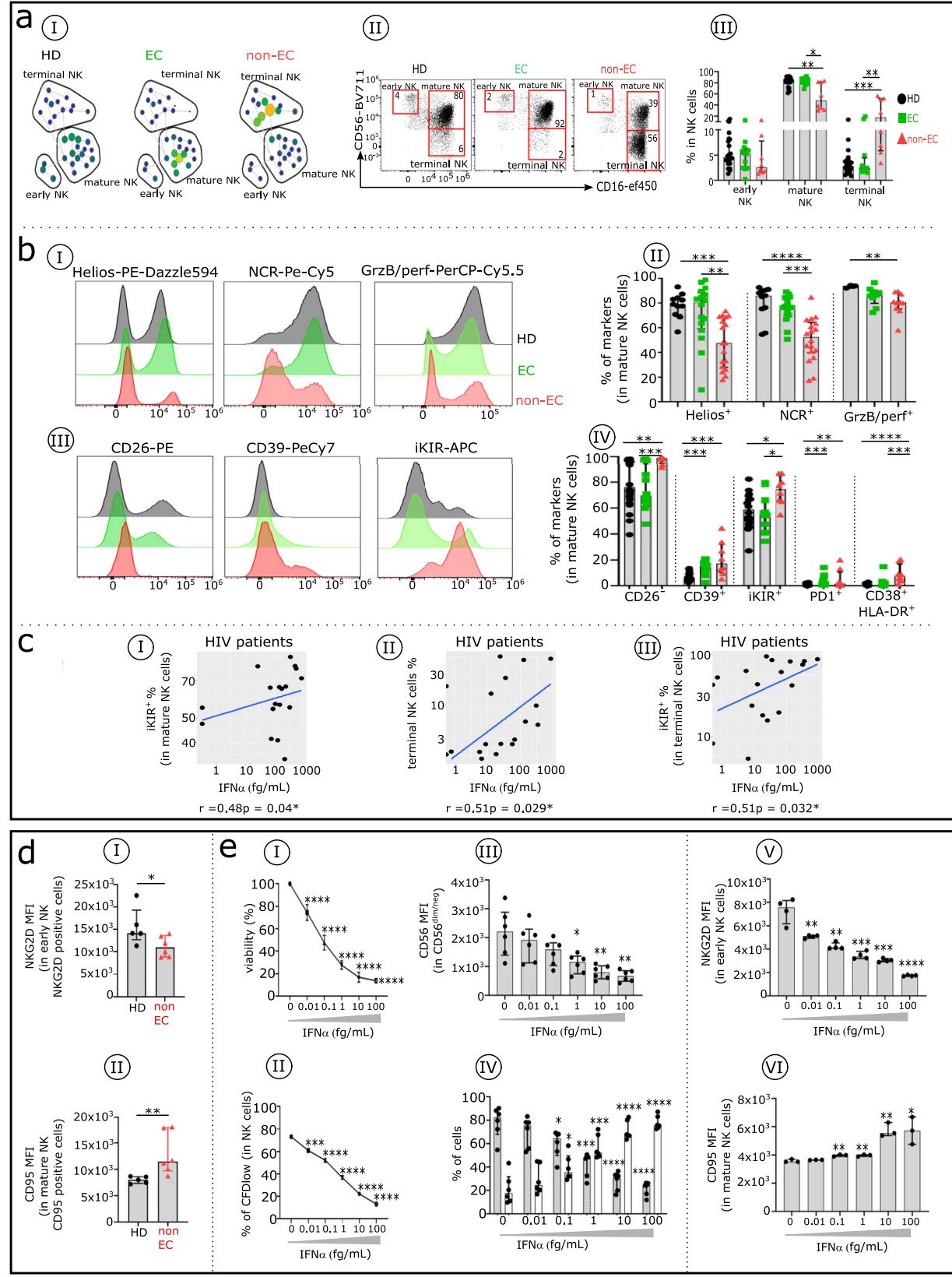

**Elevated IFNα in non-ECs is associated with a high frequency of IFNα-induced phenotypic abnormalities in CD4⁺ and CD8⁺T cells**

We first identify the major circulating CD4⁺Tconv (T helper Foxp3⁻) cell subsets at different maturation stages based on their surface expression of CCR7 and CD45RA[36]. SPADE analysis done on conventional T cells (Tconv; Fig. 3aI) and histograms (Fig. 3aII) show a varied frequency of their subtypes in both non-ECs and ECs compared with HDs. Both ECs and non-ECs have significantly lower frequencies of naive CD4⁺T cells compared with HDs. ECs have higher frequencies of CMs and effector memory (EM) T cells that re-express CD45RA (TEMRAs), and non-ECs have higher

**Fig. 2 | Comparative distribution and immune phenotypic analysis of innate immune NK cells in HDs, ECs, and non-ECs. a** Comparative analysis of NK cell subsets distribution in HDs, ECs, and non-ECs. **a1** SPADE tree with the distribution of the three main NK cell subsets in HDs (black), ECs (green), and non-ECs (red), based on CD56 and CD16 expression levels. Nodes are colored by count. **aII** Representative flow cytometry plots of NK cell subsets gated on CD19$^-$CD14$^-$TCRγδ$^-$CD3$^-$HLA-DR$^-$ cells from the three studied groups: early NK (CD56$^{bright}$/CD16$^-$), mature NK (CD56$^{dim}$/CD16$^+$), and terminal NK (CD56$^-$CD16$^+$). **aIII** Frequency of early, mature, and terminal NK cells in each studied group (HDs $n = 22$, ECs $n = 12$, and non-ECs $n = 8$). **b** Differential expression of checkpoint molecules on mature NK cells among HDs, ECs, and non-ECs. Profiles display the expression level of Helios, natural cytotoxicity receptors (NKp30, NKp44, and NKp46), granzyme/perforin (GrzB/perf) (**bI**) and CD26, CD39, inhibitory killer Ig-like receptors (iKIR) (**bIII**) on mature NK cells from HDs (black), ECs (green), and non-ECs (red). (**bII** and **bIV**). Boxplots displaying the frequency of the indicated markers in mature NK cells in each studied group. Analysis was done in 22 HDs, 12 ECs, and 8 non-ECs for all markers except for Helios and NCR (11 HDs,

16 ECs, 18 non-ECs) and GrzB/perf (3 HDs, 9 ECs, 10 non-ECs). **c** Scatterplots of the relationships between IFNα serum level and the frequencies of selected NK cell subsets. Correlations were evaluated using Spearman's rank correlation test. **d** Comparison of NKG2D et CD95 level between HDs and non-ECs in early and mature NK cells. Histograms show the expression level (measured by median fluorescence intensity [MFI]) of NKGD2 in early NK cells (**dI**) and CD95 in mature NK cells (**dII**) in non-ECs ($n = 6$) and HDs ($n = 5$). **e** IFNα effects on NK cell viability, proliferation and phenotype. Percentage of viable NK cells ($n = 3$) (**eI**) and frequency of CFD$^{low}$ NK cells ($n = 3$) (**eII**) after 7 days of culture in the presence of increasing doses of IFNα. Histograms show the expression level of CD56 in CD56$^{dim/neg}$ NK cells ($n = 6$) (**eIII**), distribution of mature (gray) and terminal NK cells (white) ($n = 6$) (**eIV**), expression levels of NKG2D in early NK cells ($n = 3$) (**eV**), and CD95 in mature NK cells ($n = 3$) (**eVI**) after 3 days of culture in the presence of IFNα. Comparisons between the two groups were performed using the Mann–Whitney *U*-test. Multiple group comparisons were made using the Kruskal–Wallis test with Dunn's multiple comparison testing. *$p < 0.05$, **$p < 0.01$, ***$p < 0.001$, and ****$p < 0.0001$. Error bars on graphs represent interquartile ranges.

## Table 1 | Contribution of immune mechanisms to EC status

| Group | Sample | VL | CD4 count (copies/mL) | IFNα (fg/mL) | IFNλ (fg/mL) | pDC (%) | CD8$^+$ CM (%) | CTL PD1$^-$ (%)† | CD8$^+$supp PD1$^-$ (%)† | Mature NK NKG2C$^+$ (%)† | Mature NK iKIR$^-$ (%)† | Mature NK iKIR$^+$ PD1$^-$ (%)† |
|---|---|---|---|---|---|---|---|---|---|---|---|---|
| B57- | EC 47 | <20 | 891 | 3.37** | 171.96 | NA | 5.7 | 37.75 | 23.15 | 11.7 | 26.8 | 72.72 |
| | EC 58 | 59 | 1745 | 5.58** | 2389.62 | 13.9 | 6.04 | 35.36 | 22.62 | 26.5 | 37.1 | 62.14 |
| | EC13 | <40 | 864 | 6.45** | 257.48 | 58.7* | 13.5* | 55.07* | 4.51 | 33.8 | 66.8* | 32.91* |
| | EC 32 | <48 | 643 | 7.66** | 319.75 | NA | NA | NA | NA | NA | NA | NA |
| | **EC31** | <48 | 587 | 67.31 | 500.06 | 28.6 | 3.86 | 47.04 | 32.25* | 23.4 | 38.7 | 59.59 |
| | EC 55 | 50 | 482 | 72.21 | 578.54 | NA | 6.91 | 29.93 | 30.19* | 23.5 | 27.7 | 72.06 |
| | EC 3 | 334 | 1140 | 74.43 | 766.29 | NA | 0.77 | 44.41 | 21.57* | 25.9 | 46.2 | 53.41 |
| B57+ | EC 63 | <40 | 632 | 0** | 338.45 | 50.4* | 27.7* | 61.37* | 8.86 | 17 | 55.8* | 43.85 |
| | EC 9 | <40 | 496 | 0** | 182.96 | 9.98 | 16.7* | 33.33 | 2.83 | 7.79 | 74.4* | 25.30* |
| | **EC65** | <48 | 1792 | 0.75** | 308.57 | 37.9 | 9.25* | 57.82* | 14.41 | 19.4 | 20.3 | 78.37 |
| | EC 6 | <75 | 1889 | 8.48** | 1875.86 | 11.5 | 21.7* | 46.04 | 7.91 | 12.8 | 61.2* | 38.60 |
| | EC 42 | <220 | 731 | 12.44** | 142.63 | 35.9 | 10.4* | 42.67 | 20.62 | 14.7 | 44.7 | 52.04 |
| | EC51 | <40 | 1250 | 18.1** | 32441.4* | 51.3* | 9.39* | 53.55* | 8.61 | 9.14 | 54.5* | 45.34 |
| | EC 4 | <40 | 952 | 25.28** | 774.38 | 8.17 | 9.37* | 25.74 | 5.94 | 5.61 | 47.2 | 52.54 |
| | **EC8** | <40 | 1018 | 57.14** | 2873.66* | 36.9 | 10.1* | 46.45 | 8.23 | 4.89 | 66.8* | 32.96* |
| | EC11$^+$ | <40 | 917 | 348.08* | 1639.51 | 7.29 | 13.9* | 17.16 | 0.95 | 9.14 | 41.3 | 51.48 |
| | EC52$^+$ | <20 | 340 | 796.94* | 2384.02* | NA | 6.85 | 38.43 | 13.71 | 12.6 | 41.7 | 57.82 |
| | EC68$^+$ | 169 | 584 | 99.63 | 830.17 | NA | NA | NA | NA | NA | NA | NA |
| B57− $n = 7$, median | | 59 | 864 | 72.21 | 500.06 | 21.25 | 5.7 | 37.75 | 22.62 | 24.7 | 37.1 | 62.14 |
| B57+ $n = 11$, median | | <40 | 917 | 99.63 | 556.415 | 11.5 | 6.85 | 38.43 | 8.42 | 10.87 | 41.7 | 51.76 |
| Non-ECs $n = 26$, median | | 21701 | 404.5 | 93.01 | 1027.92 | 13.7 | 4.155 | 42.90 | 19.12 | 28.85 | 36.4 | 59.60 |
| HDs $n = 65$, median | | NA | NA | 17.72 | 595.35 | 26.7 | 9.92 | 41.47 | 20.27 | 4.14 | 50.95 | 48.85 |
| ECs $n = 18$, median | | 114 | 877.5 | 73.32 | 500.06 | 13.9 | 5.87 | 38.09 | 8.61 | 13.75 | 40 | 53.41 |

The table details enhanced (*) or diminished (**) immune mechanisms contributing to the EC status in EC individuals compared with non-ECs.

1) Three patients lost their EC status: EC31 and EC65 were volunteers for a cART trial, and EC8 had colonic carcinoma complications (bold).

2) Three individuals (EC11, EC52, and EC68) had a serum IFNα over the non-ECs median (+).

3) Note that cytotoxic NK cells and CTL and CD8$^+$supp cells in non-ECs are phenotypically hampered (†).

*NA* not available.

frequencies of EM and TEMRA cells (Fig. 3aII). Our exploration of changes in activation/exhaustion phenotypes in CD4$^+$T cell subsets in non-ECs compared with ECs and HDs shows that the absence (CD26, CD25, CD28) or increase (CD38, HLA-DR, CD39, PD1, and CTLA-4) of these markers is linked to an exhaustion-like state associated with lack of function[22]. Each Tconv subset displays a distinct, altered phenotypic profile in non-ECs

compared with ECs and HDs (Fig. 3aIII and Supplementary Fig. 3). CM, EM, and TEMRA cells from non-ECs exhibit significantly higher phenotypic alteration scores (see methods) compared with the same cell types from ECs (Fig. 3aIV). These phenotypic abnormalities correspond in large part to the IFNα effects on CD4$^+$T cells in culture (Supplementary Table 3 and Supplementary Fig. 4). This pattern is consistent with the presence of

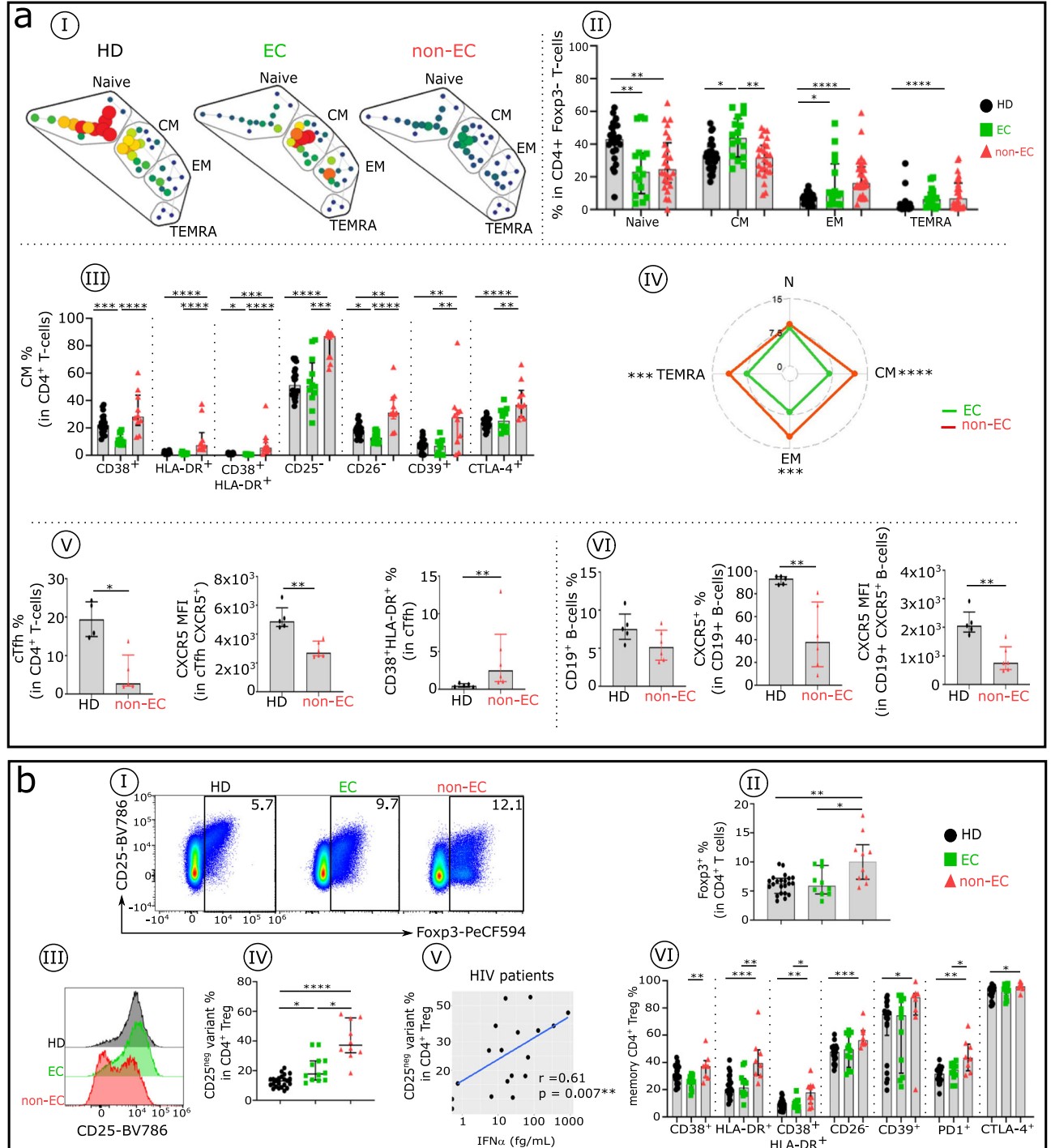

**Fig. 3 | Phenotypic analysis of CD4⁺ Tconv cell subsets, CD4⁺ Treg cell subsets and CD19⁺ B cells in HDs, ECs, and non-ECs.** Comparative immune phenotypic analysis of **a** CD4+Tconv and CD19+B cells and **b** CD4+Treg cells in non-ECs, ECs, and HDs. **aI** SPADE tree with the distribution of CD4⁺Tconv subsets in HDs (black), ECs (green), and non-ECs (red). Nodes are colored by count. CD4⁺Tconv cells can be classified into four major subsets by their expression of CD45RA and the chemokine receptor CCR7: naïve (CCR7⁺CD45RA⁺), CM (CCR7⁺CD45RA⁻), EM (CCR7⁻CD45RA⁻), and TEMRA (CCR7⁻CD45RA⁺). **aII** Frequency of naïve, CM, EM, and TEMRA cells in each studied group (HDs $n = 24$, ECs $n = 16$, and non-ECs $n = 23$). Boxplots show the expression of the indicated marker in CD4⁺ CM cells (**aIII**) across the groups (HDs $n = 22$, ECs $n = 12$, and non-ECs $n = 8$). **aIV** Radar chart of the composite scores of phenotypic cell alteration calculated for each CD4⁺Tconv subpopulation in non-ECs and ECs (see Methods). **aV** Frequency of circulating T follicular helper cells (cTfh), expression levels of CXCR5 in cTfh, and frequency of cTfh co-expressing CD38 and HLA-DR in non-ECs ($n = 6$) and HDs

($n = 5$). **aVI** Proportion of CD19⁺ B cells and frequency and expression level of CXCR5 in CD19⁺B cells in non-ECs ($n = 6$) and HDs ($n = 5$). **bI** Representative flow cytometry plots of CD25⁺Foxp3⁺ cells within CD4⁺T cells isolated from HDs ($n = 22$), ECs ($n = 12$), and non-ECs ($n = 8$). **bII** Histograms with the frequency of Foxp3 in CD4⁺T cells and **bIII** displaying the CD25 expression level in CD4⁺ Foxp3⁺T cells and **bIV** the regulatory T cell (Treg) CD25⁻ variant frequency in CD4⁺Foxp3 T cells in each studied group. **bV** Scatterplots of the relationships between frequency of the Treg CD25⁻ variant in HIV-infected patients and serum IFNα levels. **bVI** Proportion of a specific functional signaling checkpoint on memory CD4⁺ Treg (CD4⁺ Foxp3⁺CD25⁺CD45RA⁻) cells of each studied group (HDs $n = 22$, ECs $n = 12$, and non-ECs $n = 8$). Correlations were evaluated using Spearman's rank correlation test. Comparisons between the two groups were made with the Mann–Whitney $U$-test. Multiple group comparisons were assessed using the Kruskal–Wallis test with Dunn's multiple comparison testing. *$p < 0.05$, **$p < 0.01$, ***$p < 0.001$, ****$p < 0.0001$. Error bars on graphs represent interquartile ranges.

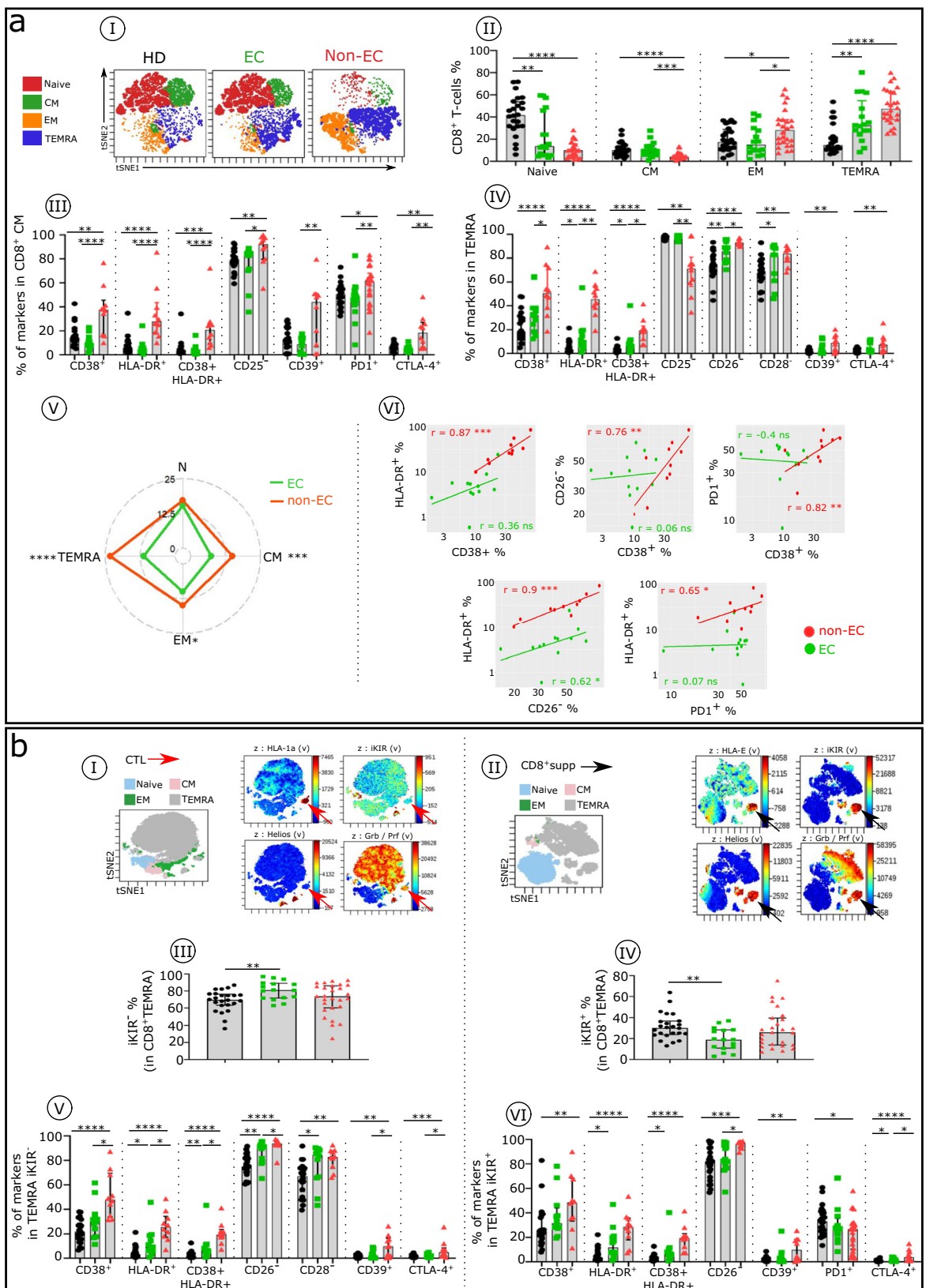

one key HIV pathogenic mediator: the effect of elevated IFNα in non-ECs. Furthermore, we find positive correlations between expression levels of different abnormally expressed markers studied in the CD4+CM cells in non-ECs, but we do not observe these correlations in ECs (Supplementary Fig. 3d).

We then investigated the circulating CXCR5+CD45RO+CD4+ T follicular helper cells (cTfh) that govern B cell hypersomatic mutation and Ig isotype commutation in follicular lymph nodes. Compared with HDs, non-ECs have a lower frequency of cTfh cells, and on these cells, they have a lower CXCR5 expression level and higher coexpression of CD38 and HLA-DR

**Fig. 4 | Comparative immune phenotypic analysis of CD8⁺T cell subsets in non-ECs, ECs, and HDs. a** Comparative analysis of CD8⁺T cell subsets distribution and phenotype in HDs, ECs, and non-ECs. **aI** Representative viSNE plot showing the distribution of CD8⁺T cell subsets, as described in Fig. 3 for CD4⁺Tconv in HDs (black), ECs (green), and non-ECs (red). **aII** Histograms of the frequencies of naïve, CM, EM, and TEMRA CD8⁺T cell subsets in each studied group (HDs $n = 24$, ECs $n = 16$, and non-ECs $n = 23$). Boxplots show the expression of the indicated marker in CD8⁺CM (**aIII**) and TEMRA (**aIV**) cells across the groups (HDs $n = 22$, ECs $n = 12$, and non-ECs $n = 8$). **aV** Radar chart of the composite scores of phenotypic cell alterations calculated for each CD8⁺T cell subpopulation in ECs and non-ECs (ECs $n = 12$ and non-ECs $n = 8$). **aVI** Scatterplots of the relationships between the expression level of indicated markers in CD8⁺CM cells (ECs $n = 12$ and non-ECs $n = 8$). **b** viSNE plot of the phenotypic difference between CD8⁺CTL (TEMRA iKIR⁻) (**bI**) and CD8+ suppressive T cells (CD8⁺supp) (TEMRA iKIR⁺) (**bII**). t-SNE plot of CD8⁺T cell subsets indicated in different colors, with viSNE projections of expression of indicated markers. Red and black arrows indicate HLA-1a–restricted and HLA-E–restricted CD8⁺supp cells, respectively. Histograms show the frequency of CD8⁺TEMRA iKIR⁻ (**bIII**) and TEMRA iKIR⁺ (**bIV**) in each studied group (HDs $n = 24$, ECs $n = 16$, and non-ECs $n = 26$). Boxplots give the proportion of specific markers on CD8⁺TEMRA iKIR⁻ (**bV**) and TEMRA iKIR⁺ (**bVI**) in each studied group (HDs $n = 22$, ECs $n = 12$, and non-ECs $n = 8$). Correlations were evaluated using Spearman's rank correlation test. Multiple group comparisons were made using the Kruskal–Wallis test with Dunn's multiple comparison testing. $*p < 0.05$, $**p < 0.01$, $***p < 0.001$, $****p < 0.0001$. Error bars on graphs represent interquartile ranges.

(Fig. 3aV). In addition, a similar frequency of CD19⁺B cells is seen between HDs and non-ECs, but non-ECs have a reduced percentage of CXCR5⁺B cells, which in turn have lower expression of CXCR5 (Fig. 3aVI). As for the regulatory CD4⁺T cell subset (CD4⁺Foxp3⁺), their frequency and function are altered in non-ECs but not in ECs (Fig. 3b). Their proportion is increased in non-ECs (Fig. 3bI, bII), and the percentage of non-functional CD25⁻ regulatory T cell (Treg) variants[19] is enhanced (Fig. 3bIII, bIV). Once again, serum IFNα level significantly correlates with this alteration (Fig. 3bV). In addition, multiple memory CD4⁺Treg phenotypic abnormalities are associated with IFNα in non-ECs, but these abnormalities are less numerous in ECs (Fig. 3bVI).

In a similar analysis of CD8⁺T cells in non-ECs compared with ECs and HDs, we examined four well-defined CD8⁺T cell subsets as described above for the Tconv. We find distinct distributions of CD8⁺T cell subsets in the three groups, as viSNE analysis (Fig. 4aI) and histograms (Fig. 4aII) show. Both ECs and non-ECs have a significantly lower frequency of naive CD4⁺T cells compared with HDs. Non-ECs also have fewer CM cells and more EM and TEMRA cells than do HDs and ECs (Fig. 4aI, aII). As for the Tconv subsets, each CD8⁺T cell subset has a distinct pattern of phenotypic alterations in non-ECs compared with ECs and HDs (Fig. 4aIII,4aIV). Non-EC CM and TEMRA cells have a higher phenotypic alteration score than these cells do in ECs (Fig. 4aV). IFNα induces a large majority of these inhibitory checkpoint receptors (Supplementary Table 3). Furthermore, as described in Fig. 4aVI, in non-ECs, we find positive correlations between expression levels of the various studied markers in CD8⁺CM cells. CD8⁺ HIV-specific cytotoxic T cells consist of effector HLA-1a–restricted CTLs and HLA-E-restricted CD8⁺supps carrying iKIRs, the human analog of Ly49, which characterize murine CD8⁺supp cells[37,38]. HLA-1a-(B)– and HLA-1b-(E)–presenting HIV peptide tetramers recognize CTLs (Fig. 4bI) and CD8⁺supps (Fig. 4bII), respectively. Using this pattern, we distinguish and phenotypically characterize the two types of cytotoxic CD8⁺T cells: CTLs (CD8⁺, TEMRA, GrzB/perf⁺, iKIR⁻, Helios⁻) and CD8⁺supps (CD8⁺, TEMRA, GrzB/perf⁺, iKIR⁺, Helios⁺). We find similar CTL frequencies in ECs and non-ECs (81.05 vs 74.15%) (Fig. 4bIII), but the inhibitory checkpoint receptor level is significantly higher in non-ECs than in ECs and HDs (Fig. 4bV), in keeping with the effect of elevated IFNα. This change leads to greater CTL activity in ECs. For the CD8⁺supp cells, non-ECs have a slightly higher percentage of these cells than do ECs (25.85 vs 18.95%) (Fig. 4bIV), with a more pronounced exhaustion-like stage (CD38, CD39, HLA-DR) or downregulated markers (CD26) (Fig. 4bVI), all of which IFNα can induce. Of note, similar to the CTL cells, the CD8⁺supp cell frequency in ECs is variable from one individual to another.

**Minimal anti-HIV immune cell phenotypic defects and control of HIV replication in ECs is linked to control of IFNα**

PCA analysis (Fig. 5aI) and heatmaps (Fig. 5aII) show that ECs, whether expressing the HLA_B57 (EC_B57+) allele or not (EC_B57−), have a distinct blood immune cell profile that is variable from individual to individual. Overall, this group expresses a few phenotypic alterations in large part induced by elevated IFNα (compared with non-ECs; Fig. 1d).

To further identify which anti-HIV immune function that is uncommon in humans could account for control of HIV replication in each EC, we investigated which anti-HIV factor(s) that neutralize circulating virions or lyse infected CD4⁺T cells might contribute to maintenance of the EC status. Considering the frequency of elevated IFNα–induced immune cell type phenotypic anomalies in non-ECs, we examined serum IFNα levels in all patients. The median is 95 fg/mL in non-ECs and 18 fg/mL in ECs. In our US HIV cohort, 16 of 20 non-ECs compared with 3 of 18 ECs show a higher IFNα level (Table 1). Interestingly, among the three ECs expressing elevated serum IFNα, EC11 and EC52 compensate their high IFNα level by expression of NK cells activating the NKG2C receptor and contributing to early lysis of infected CD4⁺T cells. The EC68 cell sample was not available (Table 1). Regarding IFNλ, it is notable that only EC51 has a high level compared with other ECs (32000 vs 1027 fg/mL). pDC frequency is higher in ECs compared with other HIV-infected individuals (32 versus 14%, Table 1). It is unlikely that greater numbers of pDCs could be a key immune parameter leading to the EC state and control of elevated IFNα. Rather, it is likely that their increase results from better control of IFNα in the early days of infection, thereby avoiding its negative impact.

Given that NK cells act at the onset of infection and before the virus setpoint, we focused on them in both EC_B57+ and EC_B57− cases. Mature EC_B57+ NK cells (9 of 10 tested) are predominantly iKIR⁻, whereas most mature EC_B57− NK cells are iKIR⁺ (Fig. 5aII, bI). Therefore, these EC_B57+ NK cells have the functional capacity to lyse infected CD4⁺T cells, releasing circulating virions as early as the beginning of the immune response. It is notable that one EC_B57− (EC13) behaves as an EC_B57+, but the mechanism is not defined (Table 1). Interestingly, in EC_B57−, we find that in mature NK cells, the expression of NKG2C counterbalances the inhibitory iKIR signals (Fig. 5bIII) and allows these NK cells to kill infected CD4⁺T cells expressing HIV peptides in an HLA-E presentation, thus reducing the viral load in EC_B57−.

The percentage of circulating CD8⁺CM cells is relatively high in EC_B57+ (identical to the HD median: 10 vs 10%) and reduced in EC_B57−, although higher than in non-ECs (median: 6 vs 4%; Table 1) (Fig. 5bIV and Supplementary Fig. 2cV). Consequently, the frequency of CTLs derived from the differentiation of CD8⁺CM cells is lower in EC_B57− compared with EC_B57+ cases. The result is a higher frequency of CTLs than CD8⁺supps in EC_B57+ (Fig. 5bV and Supplementary Fig. 5a). In contrast, EC_B57− cases have a higher percentage of CD8⁺supp cells (Fig. 5aII and Supplementary Fig. 5b) expressing less PD1 (Fig. 5bVI), and EC_B57− (EC13) behaves again as an EC_B57+. Notably, both subsets are functionally effective only during the adaptive immune response and not relevant in the critical earliest days of infection.

Each EC phenotypic profile is distinct despite few detected immune cell alterations (Fig. 5c). The majority of these phenotypic alterations, like those in non-ECs, correspond to the direct (HLA-DR, CD38) or indirect (absence of CD26, CD25) effects of IFNα seen in T cell cultures (Supplementary Table 3 and Supplementary Fig. 4). These few but irreversible alterations might occur during the early period of HIV infection in the innate phase of the immune response. They may be augmented by HIV cytopathic effects

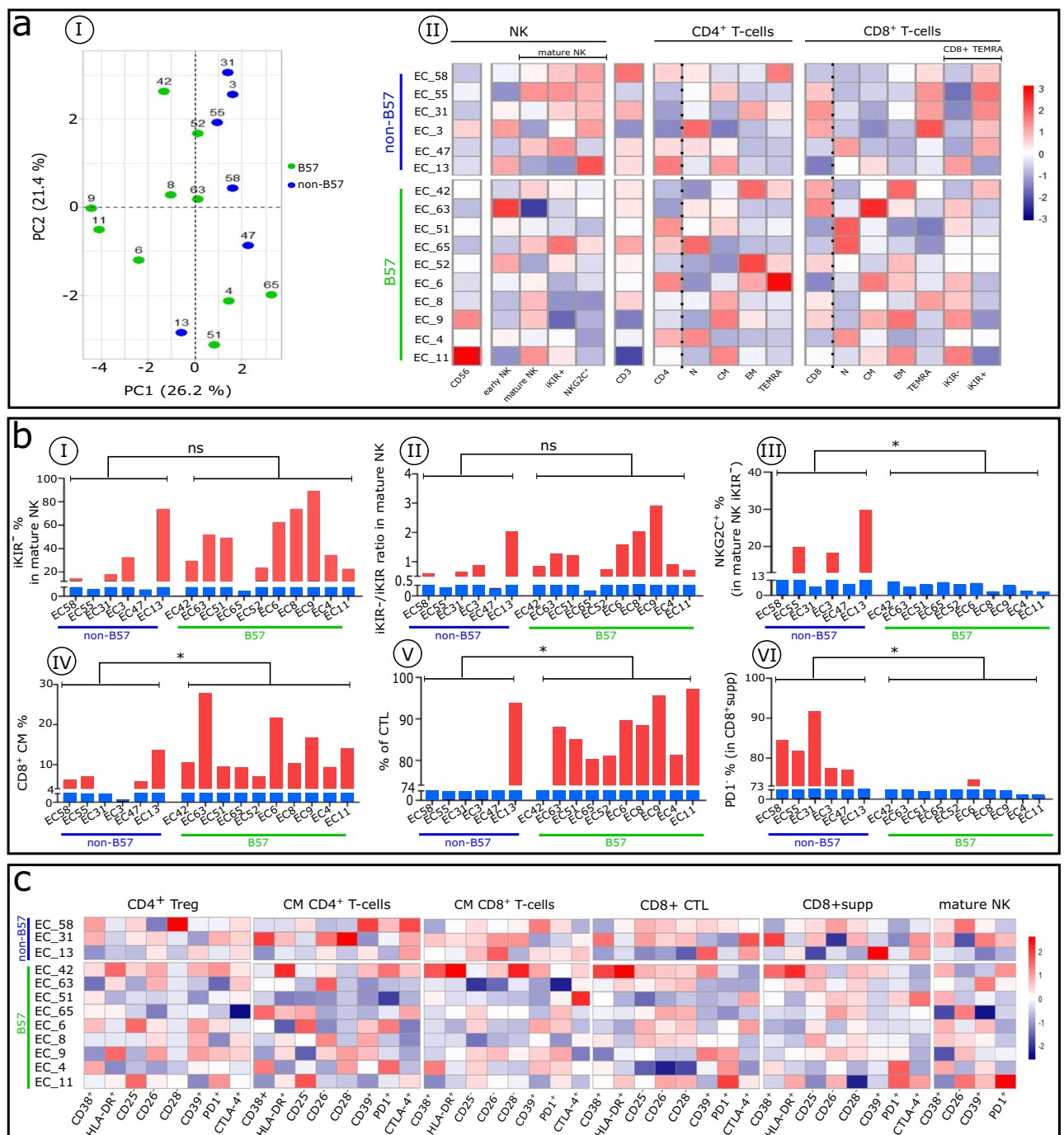

**Fig. 5 | Distinct immune cell phenotypic patterns between non-B57 and B57 ECs with an individual profile displayed by each patient in these subgroups.** **a** Comparative analysis of NK cells, CD4+ and CD8+ T-cell subset distribution in non-B57 and B57 ECs. **aI** PCA of studied non-B57 (EC_B57−) and B57 (EC_B57+) ECs based on the proportion of CD4+, CD8 + T cells, NK cells, and their subpopulations, evaluated by flow cytometry. Each point represents one participant, color-coded by group: EC_B57− (blue), EC_B57+ green). **aII** Heatmaps show the distribution of the indicated lymphocyte subsets in ECs. **b** Differential expression of checkpoint molecules on mature NK cells and different subsets of CD8+ T cells. Histograms show the frequency and index ratio of indicated subsets in mature NK (**bI–III**) and CD8+T cell compartments (**bIV–VI**). **c** Heatmaps of the frequency of indicated markers in CD4+ Treg, CD4+CM, CD8+CM, CD8+CTL, CD8+supp, and mature NK cells. Warmer colors indicate higher values and colder colors indicate lower values. Comparisons between the two groups were performed with the Mann–Whitney U-test. *$p < 0.05$, **$p < 0.01$, ***$p < 0.001$, ****$p < 0.0001$; ns not significant.

on thymic stromal cells, which occur during early infection in the innate phase of the immune response, and as reported, also are induced by IFNα[39].

## Discussion

We conclude that besides circulating HIV, the key mediator of AIDS development is elevated circulating IFNα. The main criteria for establishing

a central major mediator of pathogenesis are that it should be active in the earliest days of infection (i.e., in the innate immune phase), be elevated and circulating, and have broad pathogenic effects. The findings show that IFNα meets these criteria. In non-ECs, it circulates at elevated levels, inducing immune cell type abnormalities linked to poor immune cell function (Figs. 1–5 and Table 1). Correspondingly, in ECs, an absence of high

circulating levels of IFNα is associated with minimal immune cell type anomalies. Based on documented experimental data, we propose that ECs avoid the pathogenic vicious cycle of high HIV → high IFNα → more HIV → pathogenic IFNα. We draw this inference based on four key findings, elaborated below.

First, at infection onset, NK cells play a critical role in ECs but not in non-ECs. These cells lyse target cells before the viral setpoint and thus are likely the most effective immune factor reducing viral load and correlated IFNα levels at onset. The differences between NK cells from ECs and non-ECs (Supplementary Fig. 6) lie in their functional capacity. Non-EC NK cells expressing multiple inhibitory receptors, including iKIR and checkpoint receptors such as PD1, are likely not functional (Fig. 2bIII, IV and Table 1). Moreover, the low expression of iKIR in NK cells from $EC_{B57+}$ enables these unaltered cells to lyse infected $CD4^+T$ cells expressing $HLA_{B57}$-restricted HIV epitopes, whereas $EC_{B57-}$ NK cells expressing iKIR (Fig. 5bI, II) compensate for this inhibitory signal by expressing the activating NKG2C receptor (Fig. 5bIII), enabling lysis of HLA-E–restricted infected $CD4^+T$ cells.

The second key finding is that IFNα-induced impairment of IL-7–induced immune cell homeostasis in non-ECs results in the reduction of NK, $CCR7^+CD4^+$, and $CD8^+T$ cells. As a consequence, populations of helper and cytotoxic T cells decline during the later adaptive immune response. The third finding is that the hampered IFNα-induced initiation of the adaptive immune response in non-ECs results in diminished $CD4^+T$ cell activation and proliferation (Supplementary Table 3 and Supplementary Fig. 4). Finally, in non-ECs, inhibition of CTL and $CD8^+supp$ cell function linked to IFNα-induced inhibitory checkpoints results in loss of control of viral replication (Fig. 4 and Supplementary Table 3).

The consortium of collaborators working on the ECs has postulated a possible immune mechanism enabling control of HIV replication in ECs[2] (Fig. 5). In keeping with this concept, we find that in addition to evidence supporting our hypothesis of a putative lower infectious inoculum, each EC exhibited one or more distinct immune mechanisms (Table 1) that could contribute to the avoidance of elevated IFNα levels. These mechanisms include early lysis of infected $CD4^+T$ cells by phenotypically unaltered NK cells and during the adaptive immune response by cytotoxic CTL and $CD8^+supp$ cells. These data prompted us to consider that EC status does not depend solely on one specific genetic or immune cell profile but also on a distinct immune capacity that is peculiar to each EC case and likely additional circumstantial factors, such as a low inoculum size. There is no doubt that early contributing factors for NK lysis include HLA presentation of HIV peptides by infected cells, including the known $HLA_{B57+}$ and HLA-E, following the adaptive immune response for CTL or $CD8^+supp$ T cells. It also is clear that the $HLA_{B57}$ genotype is insufficient to account for EC status, as not all ECs are $HLA_{B57+}$ and $HLA_{B57+}$ is present in up to 11% of patients with progressive disease, similar to what is seen in the uninfected white population[4]. In keeping with Altfeld et al.[40], we suggest that HIV evasion of the cytotoxic NK cell–mediated immune response is far more deleterious for early control of viral replication than are CTL and $CD8^+supp$ cells, which are effective only during the adaptive immune response. Considering that ECs do not exhibit many immune mechanisms in non-ECs related to elevated IFNα caused by HIV disruption (Table 1), we propose that a fortuitous low infectious inoculum is an important contributor to EC status. This hypothesis involves several associated predictions now under analysis.

With a comprehensive characterization of the pathogenic levels of IFNα during both the anti-HIV innate and adaptive immune reactions, we identify elevated IFNα as the key mediator of HIV inhibition of immune cell attack in non-ECs and show how ECs naturally block HIV replication by avoiding elevated IFNα. These findings suggest that reducing IFNα production by any acceptable and efficient means could convert non-ECs to ECs. This proposal is further supported by work carried out in HIV-infected humanized mice showing that anti-IFNα treatment reduces HIV proviral DNA by 14-fold in spleen cells and 7-fold in bone marrow cells[32]. In an accompanying manuscript, we describe the mechanism of how an inefficient anti-HIV immune response promotes progressive elevation of IFNα[18].

## Data availability

Source data for the main figures in the manuscript can be accessed at https://doi.org/10.6084/m9.figshare.25125377[41].

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

## Acknowledgements
We thank the study volunteers and medical personnel. We also thank Elissa Miller for skillful administrative and editorial collaboration and Philip Embiricos for financial support sponsorship. M.M.S. was supported by R01AI147870 and IBX002358A.

## Author contributions
R.G., D.Z. and H.L.B. designed the research; H.L.B., V.S., M.M., D.B., A.H.-K. and H.S. conducted the research; H.L.B., V.S., M.M., D.B., A.H.-K., H.S., J.-D.B., A.B., R.G. and D.Z. analysed the data; S.K., M.S., C.F.L. and G.D. provided blood cells and serum samples, clinical data samples from HIV patients, and clinical input; and H.L.B., R.G. and D.Z. wrote the paper. All authors contributed to the review and editing and agreed to the published version of the manuscript.

## Competing interests
The authors declare no competing interests.
