## [Peer Review File · Communications Medicine]

Reviewers' comments:

Reviewer #1 (Remarks to the Author):

Le Buanec et al. focused on IFN- α levels measured with high-sensitivity SIMOA assays in naïve donors (HD, n=51), elite controllers (EC, n=18) and non-elite controllers (non-EC, n=26). Immunophenotyping was performed using current bioinformatics. The authors specifically examined HLA-B57, known to be associated with a higher probability of EC.

The main findings are that peripheral IFN- α levels (but not IFN- λ 2) are significantly lower in HD and EC than in non-EC, the latter correlating with changes in NK cells that contribute to an inhibitory phenotype. Treatment of purified NK cells with IFN- α at concentrations observed in vivo significantly decreased viability and proliferation as well as expression of CD56 and NKG2D in vitro. In addition, expression of CD95 was increased in a dose-dependent manner, replicating the NK cell changes observed in non-EC. PDC frequency was (significantly?) higher in EC compared with non-EC.

The number and phenotype of CD4+ and CD8+ T cells differed in non-EC compared with EC, with a more exhausted phenotype in non-EC. CCR7 expression was reduced in CD4+ T-cells from non-EC but was maintained in HLA-B57+ EC. Accordingly, exposure of CD4+ T cells to IFN- α in vitro decreased proliferation and viability. The number of Treg cells and phenotypic markers of activation/exhaustion and inhibitory checkpoints were increased in non-EC compared with EC, which was also observed for CD8+ EM and TEMRA. The percentage of iKIR- CD8+CTL was significantly increased and of iKIR+ CD8+supp significantly reduced in EC compared with HD. The latter showed a more exhausted phenotype in non-EC.

Finally, the authors compared HLA-B57+ and HLA-B57- EC. The former had higher IFN- α levels compared to the latter, but both were lower than in non-EC. HLA-B57+ EC had more iKIR- NK cells, which expressed less NKG2D and should be able to lyse CD4+ T cells presenting HIV-peptides by HLA-E. HLA-B57+ EC had more CD8+ CM and more CTL, whereas HLA-B57- EC had more CD8+supp that expressed less PD-1.

The authors conclude from their data that early innate immune responses, particularly by iKIR NK cells, are important for control at the onset of HIV-1 infection. Their conclusions are strongly supported by the changes observed in NK cells in vitro upon exposure to physiological IFN- α concentrations. Adaptive immune responses, in particular by less exhausted and less inhibited CD8+ T cells, contribute to ongoing HIV-1 control via HLA-E based recognition of infected CD4+ cells.

Major points:

- Fig. 5B and Table 1: It is unclear which differences between B57+ and B57- EC are significant. Please perform statistics.
- Extended data Fig. 1: Why were IFN α concentrations in the ng/ml range necessary to trigger the changes (in contrast to fg/ml concentrations in Fig. 2E)?
- Please provide the information whether EC and non-EC were untreated at the time of sampling.
- Statistics: which post-hoc test was used to control for multiple testing?

- Discussion: As in the accompanying manuscript, the authors speculate that the EC were coincidentally infected with a low viral load that is controlled by a rapid innate immune response, in particular by NK cells. I am not convinced that the manuscript(s) provide sufficient data to confirm this hypothesis. To find out whether the level of initial viral load truly predicts elite control, it may be helpful to test seroconverters.

- Other groups have previously published that elevated IFN- α levels play an important role in the immunopathogenesis of HIV-1 infection. The authors cite Gene Shearer's group (ref. 41 and ref. 53), they should also cite Jay Levy's group (PMID 16278001, 22203858).

Minor points:

- Fig. 1D5: Please specify whether the DC are PDC or MDC.

- Fig. 2E: Please specify on how many donors the studies using increasing IFN-alpha concentrations were performed.

- Please explain abbreviations in the figures and table legends.

- Methods: please provide reference numbers for ethical approval.

- P.15: Altfeld instead of Altflod

- P. 18: was the "Composite cell phenotypic alteration score" used in Fig. 5D?

- Ref. 17: is the journal name correct?

Reviewer #2 (Remarks to the Author):

Buanec and colleagues propose that IFN alpha is a key determinant for HIV pathogenicity. The authors present numerous lines of evidence that IFN alpha undermines NK and other killer cell function. The results are provocative and, if correct, important to our understanding of HIV immunopathogenicity. My main concern lies with the nature of the cohorts. Do the authors have information of IFN alpha levels in the EC and non-EC subjects either at acute viremia or at set point? It would be interesting to know whether IFN levels at either of those episodes might predict subsequent disease sequelae. Furthermore, a discussion of treatment interruption is warranted and whether IFN levels at interruption reflect the rate of progression of the subjects prior to initiating therapy.

Answer to Paper 1 reviewers' comments

Reviewer #1 (Remarks to the Author):

Le Buanec et al. focused on IFN- α levels measured with high-sensitivity SIMOA assays in naïve donors (HD, n=51), elite controllers (EC, n=18) and non-elite controllers (non-EC, n=26). Immunophenotyping was performed using current bioinformatics. The authors specifically examined HLA-B57, known to be associated with a higher probability of EC.

The main findings are that peripheral IFN- α levels (but not IFN- λ 2) are significantly lower in HD and EC than in non-EC, the latter correlating with changes in NK cells that contribute to an inhibitory phenotype. Treatment of purified NK cells with IFN- α at concentrations observed in vivo significantly decreased viability and proliferation as well as expression of CD56 and NKG2D in vitro. In addition, expression of CD95 was increased in a dose-dependent manner, replicating the NK cell changes observed in non-EC. PDC frequency was (significantly?) higher in EC compared with non-EC.

The number and phenotype of CD4+ and CD8+ T cells differed in non-EC compared with EC, with a more exhausted phenotype in non-EC. CCR7 expression was reduced in CD4+ T-cells from non-EC but was maintained in HLA-B57+ EC. Accordingly, exposure of CD4+ T cells to IFN- α in vitro decreased proliferation and viability. The number of Treg cells and phenotypic markers of activation/exhaustion and inhibitory checkpoints were increased in non-EC compared with EC, which was also observed for CD8+ EM and TEMRA. The percentage of iKIR- CD8+CTL was significantly increased and of iKIR+ CD8+supp significantly reduced in EC compared with HD. The latter showed a more exhausted phenotype in non-EC.

Finally, the authors compared HLA-B57+ and HLA-B57- EC. The former had higher IFN- α levels compared to the latter, but both were lower than in non-EC. HLA-B57+ EC had more iKIR- NK cells, which expressed less NKG2D and should be able to lyse CD4+ T cells presenting HIV-peptides by HLA-E. HLA-B57+ EC had more CD8+ CM and more CTL, whereas HLA-B57- EC had more CD8+supp that expressed less PD-1.

The authors conclude from their data that early innate immune responses, particularly by iKIR NK cells, are important for control at the onset of HIV-1 infection. Their conclusions are strongly supported by the changes observed in NK cells in vitro upon exposure to physiological IFN- α concentrations. Adaptive immune responses, in particular by less exhausted and less inhibited CD8+ T cells, contribute to ongoing HIV-1 control via HLA-E based recognition of infected CD4+ cells.

Major points:

- *Fig. 5B and Table 1: It is unclear which differences between B57+ and B57- EC are significant. Please perform statistics.*

Response: In the revised manuscript, we have indicated the statistics on graphs of the Fig. 5B. Concerning Table 1, its purpose is to show in EC patients their heterogeneous specific immunological characteristics/immune mechanisms on an individual scale, potentially contributing to their status. That's why we didn't attempt any statistics. They are distinct and that is the very point. They are not impacted by the pathogenic effect of a single key mediator, elevated IFN α . We have tried to make this now clear in the discussion. We additionally point the comparative median values of each characteristic in order to show the differences between non-EC, ECB57+, EC57- and HD. Most importantly, this referee's question is important for the reader. Indeed, our answer stresses that EC don't not possess unique genetic capacity including the known HLA-B57+/HLA-B57- or other HLA ones but they do express cytotoxic NK, CTL and CD8+supp cells distinct profiles from one to another individual EC, as we showed in Fig. 5. This finding prompted us to hypothesize the evident contribution of the infectious inoculum size not yet reported to our knowledge as one among other factors leading to EC status.

- *Extended data Fig. 1: Why were IFN α concentrations in the ng/ml range necessary to trigger the changes (in contrast to fg/ml concentrations in Fig. 2E)?*

Response: We thank the reviewer for this question, its answer included in the revised manuscript clarifies that IFN α levels depend on its source (tissue or blood). Indeed, as other cytokines, IFN α has short half-life (Shechter Y, et al. Prolonging the half-life of human interferon-alpha 2 in circulation: Design, preparation, and analysis of (2-sulfo-9-fluorenylmethoxycarbonyl)-interferon-alpha 2. Proc Natl Acad Sci U S A. 2001 Jan 30;98(3):1212-7. doi: 10.1073/pnas.98.3.1212. PMID: 11158619; PMCID: PMC14734), and acts in tissues and particularly in lymphoid organs locally in a juxtacrine or

paracrine manner and is released into the body fluid at only background level (< 100 fg/ml in HD but also in most of EC). However, when abnormally highly produced in lymph nodes, as in non-EC, IFN α abnormally overflows into the body fluid and becomes diluted in serum (at concentration 10-1000 fg/mL).

We appreciate you bringing it up because it means that readers would not have known what we are really getting at.

During acute HIV infection, IFN α production by pDC in lymph nodes, contributes to the early antiviral protection by induction of an ISG-based cellular antiviral program. The level of IFN α produced is correlated with viral load. If the infection with HIV occurs in the presence of a high dose of virus, subsequently there will be a production of elevated IFN α in the tissues which abnormally only overflow in the body fluid. This residual IFN α , diluted in the body fluid, is present at low concentrations (10-1000 fg/ml) in the serum of patients and not in serum of HD and most EC. By contrast, elevated concentration of IFN α used in CD4⁺ T-cells cultures correspond to the one present in tissues and not to the one abnormally overflowing in body fluid. The later obviously is much lower concentrated than the one present in tissues.

In the introduction of the revised text, in the paragraph 2, we modify the last line as follows: "Consequently, in inflammatory HIV lymphoid tissues, elevated IFN α locally hampers CD4⁺ T helper and CD4⁺ follicular T helper cells function and thus inhibit the initiation phase of the anti-HIV adaptive IR, leading notably to the development of HIV-specific neutralizing antibodies responses. Noteworthy even at a high level, IFN λ does not hamper CD4⁺ T-cells-induced initiation of the IR (extended data Fig. 1)."

- Please provide the information whether EC and non-EC were untreated at the time of sampling.

Response : The EC were not treated at the time of sampling. For non-EC patients, all samples were collected when patients were not yet on treatment except for patients AZT 71374 and AZT 72336 who had been on treatment for a few days.

We have indicated this information in the revised text in the table giving the clinical information of the patients (Extended data Table 2).

- Statistics: which post-hoc test was used to control for multiple testing?

Response : Comparisons between groups were performed with unpaired nonparametric Mann-Whitney as indicated in the statistical analyses section of the material and methods of the manuscript.

In the revised manuscript, comparisons between two groups are performed with unpaired nonparametric Mann-Whitney and comparisons among multiple groups are performed using Kruskal–Wallis test followed by Dunn’s Post-hoc test. Methods of analysis were also indicated in the statistical analyses section of the material and methods in the revised text and in the figure legends.

- Discussion: As in the accompanying manuscript, the authors speculate that the EC were coincidentally infected with a low viral load that is controlled by a rapid innate immune response, in particular by NK cells. I am not convinced that the manuscript(s) provide sufficient data to confirm this hypothesis. To find out whether the level of initial viral load truly predicts elite control, it may be helpful to test seroconverters.

Response: Our speculation that EC status could in part depend on the size of the HIV inoculum is based on our serologic and cellular experimental data reported here and also in our companion manuscript, and, please, note, we present this as a hypothesis. It was not meant to be solved in this report and it is noted that it is a hypothesis which indeed we are testing, and results so far support our hypothesis. However, once again, answering conclusively this hypothesis is triggered by these following items:

- 1) Lack of an exclusive HLA genetic background,
- 2) Heterogeneous cytotoxic capacity of NK, CTL and CD8⁺ supp cells, variable between each EC observed in this study (text and Fig. 5).
- 3) By other several studies supporting our hypothesis, which include:

a) Absence of EC in hemophiliacs infected with HIV when they were inoculated repeatedly with Factor VII from plasma contaminated with high level of HIV during 1980-1985 before the development of HIV blood test. This observation was made particularly with hemophiliac's patients followed up by Prof. Alessandro Gringeri in the Hemophiliac cohort in Milan (Italy).

b) The risk of contracting HIV from an occupational exposure (nurses/ laboratory technicians) is most often very low, because people are exposed to low doses of HIV.

c) Reports show that elite controllers have a much lower viral load during primary infection and rapid control of viral replication than non-controllers. Goujard et al had the opportunity to study patients who were enrolled in the French PRIMO cohort at the time of primary HIV-1 infection and who subsequently exhibited spontaneous control of viral replication. The median HIV RNA level at enrollment was statistically significantly lower for the early controllers than it was for the noncontrollers (Goujard C et al. Spontaneous control of viral replication during primary HIV infection: when is "HIV controller" status established? Clin Infect Dis. 2009 Sep 15;49(6):982-6. doi: 10.1086/605504. PMID: 19681706). Other team also show that elite controllers exhibit lower viral loads during acute infection and at set point than non-controllers (Okulicz JF et al; Infectious Disease Clinical Research Program (IDCRP) HIV Working Group. Clinical outcomes of elite controllers, viremic controllers, and long-term nonprogressors in the US Department of Defense HIV natural history study. J Infect Dis. 2009 Dec 1;200(11):1714-23. doi: 10.1086/646609. PMID: 19852669; Woldemeskel BA et al. Viral reservoirs in elite controllers of HIV-1 infection: Implications for HIV cure strategies. EBioMedicine. 2020 Dec;62:103118. PMID: 33181459; PMCID: PMC7658501). In fact, it is the logical explanation of EC since there is a series of clinical outcomes as reviewer knows well from Fast to Slow to LTNP to EC in the progression or lack of progression of AIDS in keeping with the variation found with inoculum size by most viruses but surprisingly not previously proposed by others to explain EC.

- Other groups have previously published that elevated IFN- α levels play an important role in the immunopathogenesis of HIV-1 infection. The authors cite Gene Shearer's group (ref. 41 and ref. 53), they should also cite Jay Levy's group (PMID 16278001, 22203858).

Response : In 1982, IFN α was the first abnormal serum parameter to be identified in patients with AIDS (DeStefano E et al. Acid-labile human leukocyte interferon in homosexual men with Kaposi's sarcoma and lymphadenopathy. *J Infect Dis.* 1982 Oct;146(4):451-9. doi: 10.1093/infdis/146.4.451. PMID: 7119475). Since, the pathogenic role of elevated IFN α in AIDS has been reported in numerous papers by our group (Lachgar A et al. Involvement of α -interferon in HIV-1 induced immunosuppression. A potential target for AIDS prophylaxis and treatment. *Biomed & Pharmacother* (1994) 48:73-77; Gallo R.C. et al. Targeting Tat and IFN α as a therapeutic AIDS vaccine. *DNA and Cell Biology* (2002) 21:611-618 (2002) 21:611-618) and confirmed by other groups including Shearer's (2005, 2008) and Levy's (2005,2012) ones. Given the multiple reports on the topic, we have mentioned the first fundamental papers reported by our group together with representative references on the subject. We have also mentioned the article presenting clinical trial results from the EURIS phase 2B placebo control, carried out in 5 hospital centers from 1996-1998 on 240 patients (Gringeri A et al. Active anti-interferon- α immunization: A European-Israeli randomized, double-blind, placebo-controlled clinical trial in 242 HIV-1-infected patients (the EURIS Study). *J Acquir Immune Defic Syndr Hum Retrovirol* (1999) 20:358-370), clinical trial preceded by a phase 1b. The EURIS trial showed a biological (CD4⁺ T-cell count) and clinical (AIDS related symptoms) benefits in vaccine responders. Phase 3 was not continued due to the arrival of triple therapy (cART) at the same time. Nevertheless, to satisfy the referee 1 demand we added one reference of Levy's group in the revised manuscript. This is the reference 28, which appears in the paragraph of the results section entitled "Elevated IFN α induces an abnormal frequency and phenotypic alterations of NK-cells and other innate cell types in non-EC but not in EC".

In the present study reported in the two companion manuscripts, we show that besides HIV direct effect per se, elevated IFN α is a key mediator for HIV pathogenicity. It is not just that IFN α is enhanced in AIDS; elevated IFN α triggers multiple pathogenic effects as we show here resumed in the companion paper

extended data Table 2. Moreover, elevated IFN α not only increases the expression of CCR5, the second HIV receptor on the surface of stimulated CD4⁺ T-cells but also inhibits the production of C-C chemokines as reported earlier by our group (Zagury D. et al.. Interferon alpha and Tat involvement in the immunosuppression of uninfected T cells and C-C chemokine decline in AIDS. Proc. Natl. Acad. Sci. U. S. A. 95, 3851–3856 (1998)). These chemokines protect cells from HIV infection (Cocchi F. et al.. Identification of RANTES, MIP-1 alpha, and MIP-1 beta as the major HIV-suppressive factors produced by CD8⁺ T cells. Science. 1995 Dec 15;270(5243):1811-5) by inducing internalization of the CCR5 receptor. All these direct and indirect actions of high IFN α promote the infection of CD4⁺ T-cells and the spread of HIV in other lymphoid foci. Beside HIV1 per se, elevated IFN α hampers activation of CD4⁺ T helper and follicular T helper cells which initiate the HIV specific immune reaction leading to production of neutralizing antibodies.

If the infection occurs in the presence of a high dose of virus, subsequently there will be a correlated elevated dose of IFN α , and an IFN α -induced increase of CCR5 expression on CD4⁺ T-cells and a progression of the pathogenesis leading to AIDS.

Minor points:

- Fig. 1D5: Please specify whether the DC are pDC or mDC.

Response : Figure 1B5 shows the frequency of dendritic cells (DC) which is defined by cytometry as a Lineage neg and HLA-DR + cell population (Lin- HLA-DR+). We have specified this in the revised manuscript on the graph and in the legend of the figure.

Within DC, pDC are characterized by the CD123⁺ CD11C⁻ phenotype and mDC by the CD123⁻ CD11C⁺ phenotype, as indicated in the extended data fig. 3 which presents the gating strategies of the different cell subpopulations studied.

- Fig. 2E: Please specify on how many donors the studies using increasing IFN-alpha concentrations were performed.

Response : The number of donors in the studies using increasing IFN α concentrations is indicated in the fig legends as follows: “Percentage of viable NK-cells (n=3) (E1) and frequency of CF^Dlow NK-cells (n=3) (E2) after 7 days of culture in presence of increasing doses of IFN α . Histograms showing the expression level of CD56 in CD56^{dim}/neg NK-cells (n=6) (E3), distribution of mature (grey) and terminal NK-cells (white) (n=6) (E4), expression levels of NKG2D in early NK-cells (n=3) (E5) and CD95 in mature NK-cells (n=3) (E6) after 3 days of culture in presence of IFN α .”

- Please explain abbreviations in the figures and table legends.

Response : We removed the abbreviations in the titles of the legends and indicated in the legend what the abbreviation corresponds to when it appears the first time.

We modify the text legends in the revised manuscript as follows :

- Legend Fig.1 : Title : non-EC, EC and HD were replaced by non-Elite Controllers, Elite Controllers and Healthy Donors.
- Legend Fig.2 : NCR, GrzB/perf and iKIR were replaced by Natural cytotoxicity receptors, Granzyme/Perforin and Inhibitory Killer Ig-Like Receptors.
- In the extended tables describing the patients we have added a legend to explain the abbreviations. Dx : Diagnosis, M : Male, F : Female, AA : Afro American, IDU : Injection Drug Use, HS : Homosexual, MSM : Men who have Sex with Men, NA : Non Available

-Methods: please provide reference numbers for ethical approval.

Response : The IRB number for the study in US is H-29331. In Liege (Belgium) the number of the agreement for the study is 2020/418. We added it in the material and method in the human samples section.

- P.15: Altfeld instead of Altfléd

Response : Thank you for pointing out the typo. We fixed it in the text of the revised manuscript

- P. 18: was the “Composite cell phenotypic alteration score” used in Fig. 5D?

Response : The Composite cell phenotypic alteration score was used in the Fig. 3A4 and 4A5.

- Ref. 17: is the journal name correct?

Response : Thank you for pointing out the typo. We fixed it in the text of the revised manuscript as follows:

“Gringeri, A. et al. Active anti-interferon-alpha immunization: a European-Israeli, randomized, double-blind, placebo-controlled clinical trial in 242 HIV-1--infected patients (the EURIS study). J. Acquir. Immune Defic. Syndr. Hum. Retrovirology. 20, 358–370 (1999).”

Reviewer #2 (Remarks to the Author):

Buanec and colleagues propose that IFN alpha is a key determinant for HIV pathogenicity. The authors present numerous lines of evidence that IFN alpha undermines NK and other killer cell function. The results are provocative and, if correct, important to our understanding of HIV immunopathogenicity. My main concern lies with the nature of the cohorts. Do the authors have information of IFN alpha levels in the EC and non-EC subjects either at acute viremia or at set point? It would be interesting to know whether IFN levels at either of those episodes might predict subsequent disease sequelae. Furthermore, a discussion of treatment interruption is warranted and whether IFN levels at interruption reflect the rate of progression of the subjects prior to initiating therapy.

Response : We agree with referee 2 that it would be interesting to know whether IFN α levels at either acute viremia or at set point might predict subsequent disease sequelae.

We did not have the opportunity to follow IFN α level in Elite controller and non-controller patients from the onset of their infection. However, comparisons of SIV infection in primate species that develop AIDS-like disease and species without disease symptoms indicate that an elevated IFN α a serum concentration occur only during pathogenic infection in macaques, whereas natural SIV hosts, without disease progression, have weaker serum IFN α concentration levels (Mandl, J. N. et al. Divergent TLR7 and TLR9 signaling and type I interferon production distinguish pathogenic and nonpathogenic AIDS virus infections. *Nature Med.* 14, 1077–1087 (2008) ; Jacquelin, B. et al. Nonpathogenic SIV infection of African green monkeys induces a strong but rapidly controlled type I IFN response. *J. Clin. Invest.* 119, 3544–3555 (2009)). Similar findings have been made in individuals infected with HIV. Rapid progressors show stronger IFN α signatures than viraemic non-progressors (Rotger, M. et al. Comparative transcriptomics of extreme phenotypes of human HIV-1 infection and SIV infection in sooty mangabey and rhesus macaque. *J. Clin. Invest.* 121, 2391–2400 (2011)). These studies suggest a link between sustained serum IFN α levels and disease progression.

In 2017, Cheng et al published on the targeting of type I mediated activation in humanized mice models of HIV using IFNAR antagonists to restore immune functions and reduce HIV reservoirs. Cheng et al used an anti-IFNAR1 antibody in NRG-BLT humanized mice. cART was introduced 4 weeks after HIV1 challenge and anti-IFNAR1 was administered after seven weeks. cART was then interrupted after 12 weeks of administration. They demonstrated that blockade of the IFN receptor during the chronic phase of

infection reduces the levels of T cell activation, reduces expression of the inhibitory receptors PD-1 and TIM-3, and improves cytokine production by CD8⁺ T-cells. Above all, they show that type I IFN blockade during ART administration markedly reduced the frequency of cells harboring replication-competent HIV, the so-called “reservoir” and caused a delayed rebound of viremia after ART was discontinued (L. Cheng et al, Blocking type I interferon signaling enhances T cell recovery and reduces HIV1 reservoirs, *J. Clin. Invest.* 127 (2017) 269–279. doi:10.1172/JCI90745).

REVIEWERS' COMMENTS:

Reviewer #1 (Remarks to the Author):

I appreciate the responses of the authors and have no further comments.

Reviewer #2 (Remarks to the Author):

The authors have adequately addressed my concerns.

Answer to Paper 1 reviewers' comments

Reviewer #1 (Remarks to the Author):

Le Buanec et al. focused on IFN- α levels measured with high-sensitivity SIMOA assays in naïve donors (HD, n=51), elite controllers (EC, n=18) and non-elite controllers (non-EC, n=26). Immunophenotyping was performed using current bioinformatics. The authors specifically examined HLA-B57, known to be associated with a higher probability of EC.

The main findings are that peripheral IFN- α levels (but not IFN- λ 2) are significantly lower in HD and EC than in non-EC, the latter correlating with changes in NK cells that contribute to an inhibitory phenotype. Treatment of purified NK cells with IFN- α at concentrations observed in vivo significantly decreased viability and proliferation as well as expression of CD56 and NKG2D in vitro. In addition, expression of CD95 was increased in a dose-dependent manner, replicating the NK cell changes observed in non-EC. PDC frequency was (significantly?) higher in EC compared with non-EC.

The number and phenotype of CD4+ and CD8+ T cells differed in non-EC compared with EC, with a more exhausted phenotype in non-EC. CCR7 expression was reduced in CD4+ T-cells from non-EC but was maintained in HLA-B57+ EC. Accordingly, exposure of CD4+ T cells to IFN- α in vitro decreased proliferation and viability. The number of Treg cells and phenotypic markers of activation/exhaustion and inhibitory checkpoints were increased in non-EC compared with EC, which was also observed for CD8+ EM and TEMRA. The percentage of iKIR- CD8+CTL was significantly increased and of iKIR+ CD8+supp significantly reduced in EC compared with HD. The latter showed a more exhausted phenotype in non-EC.

Finally, the authors compared HLA-B57+ and HLA-B57- EC. The former had higher IFN- α levels compared to the latter, but both were lower than in non-EC. HLA-B57+ EC had more iKIR- NK cells, which expressed less NKG2D and should be able to lyse CD4+ T cells presenting HIV-peptides by HLA-E. HLA-B57+ EC had more CD8+ CM and more CTL, whereas HLA-B57- EC had more CD8+supp that expressed less PD-1.

The authors conclude from their data that early innate immune responses, particularly by iKIR NK cells, are important for control at the onset of HIV-1 infection. Their conclusions are strongly supported by the changes observed in NK cells in vitro upon exposure to physiological IFN- α concentrations. Adaptive immune responses, in particular by less exhausted and less inhibited CD8+ T cells, contribute to ongoing HIV-1 control via HLA-E based recognition of infected CD4+ cells.

Major points:

- Fig. 5B and Table 1: It is unclear which differences between B57+ and B57- EC are significant. Please perform statistics.

Response: In the revised manuscript, we have indicated the statistics on graphs of the Fig. 5B. Concerning Table 1, its purpose is to show in EC patients their heterogeneous specific immunological characteristics/immune mechanisms on an individual scale, potentially contributing to their status. That's why we didn't attempt any statistics. They are distinct and that is the very point. They are not impacted by the pathogenic effect of a single key mediator, elevated IFN α . We have tried to make this now clear in the discussion. We additionally point the comparative median values of each characteristic in order to show the differences between non-EC, ECB57+, EC57- and HD. Most importantly, this referee's question is important for the reader. Indeed, our answer stresses that EC don't not possess unique genetic capacity including the known HLA-B57+/HLA-B57- or other HLA ones but they do express cytotoxic NK, CTL and CD8+supp cells distinct profiles from one to another individual EC, as we showed in Fig. 5. This finding prompted us to hypothesize the evident contribution of the infectious inoculum size not yet reported to our knowledge as one among other factors leading to EC status.

- Extended data Fig. 1: Why were IFN α concentrations in the ng/ml range necessary to trigger the changes (in contrast to fg/ml concentrations in Fig. 2E)?

Response: We thank the reviewer for this question, its answer included in the revised manuscript clarifies that IFN α levels depend on its source (tissue or blood). Indeed, as other cytokines, IFN α has short half-life (Shechter Y, et al. Prolonging the half-life of human interferon-alpha 2 in circulation: Design, preparation, and analysis of (2-sulfo-9-fluorenylmethoxycarbonyl)-interferon-alpha 2. Proc Natl Acad Sci U S A. 2001 Jan 30;98(3):1212-7. doi: 10.1073/pnas.98.3.1212. PMID: 11158619; PMCID: PMC14734), and acts in tissues and particularly in lymphoid organs locally in a juxtacrine or

paracrine manner and is released into the body fluid at only background level (< 100 fg/ml in HD but also in most of EC). However, when abnormally highly produced in lymph nodes, as in non-EC, IFN α abnormally overflows into the body fluid and becomes diluted in serum (at concentration 10-1000 fg/mL).

We appreciate you bringing it up because it means that readers would not have known what we are really getting at.

During acute HIV infection, IFN α production by pDC in lymph nodes, contributes to the early antiviral protection by induction of an ISG-based cellular antiviral program. The level of IFN α produced is correlated with viral load. If the infection with HIV occurs in the presence of a high dose of virus, subsequently there will be a production of elevated IFN α in the tissues which abnormally only overflow in the body fluid. This residual IFN α , diluted in the body fluid, is present at low concentrations (10-1000 fg/ml) in the serum of patients and not in serum of HD and most EC. By contrast, elevated concentration of IFN α used in CD4⁺ T-cells cultures correspond to the one present in tissues and not to the one abnormally overflowing in body fluid. The later obviously is much lower concentrated than the one present in tissues.

In the introduction of the revised text, in the paragraph 2, we modify the last line as follows: "Consequently, in inflammatory HIV lymphoid tissues, elevated IFN α locally hampers CD4⁺ T helper and CD4⁺ follicular T helper cells function and thus inhibit the initiation phase of the anti-HIV adaptive IR, leading notably to the development of HIV-specific neutralizing antibodies responses. Noteworthy even at a high level, IFN λ does not hamper CD4⁺ T-cells-induced initiation of the IR (extended data Fig. 1)."

- Please provide the information whether EC and non-EC were untreated at the time of sampling.

Response : The EC were not treated at the time of sampling. For non-EC patients, all samples were collected when patients were not yet on treatment except for patients AZT 71374 and AZT 72336 who had been on treatment for a few days.

We have indicated this information in the revised text in the table giving the clinical information of the patients (Extended data Table 2).

- *Statistics: which post-hoc test was used to control for multiple testing?*

Response : Comparisons between groups were performed with unpaired nonparametric Mann-Whitney as indicated in the statistical analyses section of the material and methods of the manuscript.

In the revised manuscript, comparisons between two groups are performed with unpaired nonparametric Mann-Whitney and comparisons among multiple groups are performed using Kruskal–Wallis test followed by Dunn’s Post-hoc test. Methods of analysis were also indicated in the statistical analyses section of the material and methods in the revised text and in the figure legends.

- *Discussion: As in the accompanying manuscript, the authors speculate that the EC were coincidentally infected with a low viral load that is controlled by a rapid innate immune response, in particular by NK cells. I am not convinced that the manuscript(s) provide sufficient data to confirm this hypothesis. To find out whether the level of initial viral load truly predicts elite control, it may be helpful to test seroconverters.*

Response: Our speculation that EC status could in part depend on the size of the HIV inoculum is based on our serologic and cellular experimental data reported here and also in our companion manuscript, and, please, note, we present this as a hypothesis. It was not meant to be solved in this report and it is noted that it is a hypothesis which indeed we are testing, and results so far support our hypothesis. However, once again, answering conclusively this hypothesis is triggered by these following items:

- 1) Lack of an exclusive HLA genetic background,
- 2) Heterogeneous cytotoxic capacity of NK, CTL and CD8⁺ supp cells, variable between each EC observed in this study (text and Fig. 5).
- 3) By other several studies supporting our hypothesis, which include:

a) Absence of EC in hemophiliacs infected with HIV when they were inoculated repeatedly with Factor VII from plasma contaminated with high level of HIV during 1980-1985 before the development of HIV blood test. This observation was made particularly with hemophiliac's patients followed up by Prof. Alessandro Gringeri in the Hemophiliac cohort in Milan (Italy).

b) The risk of contracting HIV from an occupational exposure (nurses/ laboratory technicians) is most often very low, because people are exposed to low doses of HIV.

c) Reports show that elite controllers have a much lower viral load during primary infection and rapid control of viral replication than non-controllers. Goujard et al had the opportunity to study patients who were enrolled in the French PRIMO cohort at the time of primary HIV-1 infection and who subsequently exhibited spontaneous control of viral replication. The median HIV RNA level at enrollment was statistically significantly lower for the early controllers than it was for the noncontrollers (Goujard C et al. Spontaneous control of viral replication during primary HIV infection: when is "HIV controller" status established? Clin Infect Dis. 2009 Sep 15;49(6):982-6. doi: 10.1086/605504. PMID: 19681706). Other team also show that elite controllers exhibit lower viral loads during acute infection and at set point than non-controllers (Okulicz JF et al; Infectious Disease Clinical Research Program (IDCRP) HIV Working Group. Clinical outcomes of elite controllers, viremic controllers, and long-term nonprogressors in the US Department of Defense HIV natural history study. J Infect Dis. 2009 Dec 1;200(11):1714-23. doi: 10.1086/646609. PMID: 19852669; Woldemeskel BA et al. Viral reservoirs in elite controllers of HIV-1 infection: Implications for HIV cure strategies. EBioMedicine. 2020 Dec;62:103118. PMID: 33181459; PMCID: PMC7658501). In fact, it is the logical explanation of EC since there is a series of clinical outcomes as reviewer knows well from Fast to Slow to LTNP to EC in the progression or lack of progression of AIDS in keeping with the variation found with inoculum size by most viruses but surprisingly not previously proposed by others to explain EC.

- Other groups have previously published that elevated IFN- α levels play an important role in the immunopathogenesis of HIV-1 infection. The authors cite Gene Shearer's group (ref. 41 and ref. 53), they should also cite Jay Levy's group (PMID 16278001, 22203858).

Response : In 1982, IFN α was the first abnormal serum parameter to be identified in patients with AIDS (DeStefano E et al. Acid-labile human leukocyte interferon in homosexual men with Kaposi's sarcoma and lymphadenopathy. *J Infect Dis.* 1982 Oct;146(4):451-9. doi: 10.1093/infdis/146.4.451. PMID: 7119475). Since, the pathogenic role of elevated IFN α in AIDS has been reported in numerous papers by our group (Lachgar A et al. Involvement of α -interferon in HIV-1 induced immunosuppression. A potential target for AIDS prophylaxis and treatment. *Biomed & Pharmacother* (1994) **48**:73-77; Gallo R.C. et al. Targeting Tat and IFN α as a therapeutic AIDS vaccine. *DNA and Cell Biology* (2002) 21:611-618 (2002) 21:611-618) and confirmed by other groups including Shearer's (2005, 2008) and Levy's (2005,2012) ones. Given the multiple reports on the topic, we have mentioned the first fundamental papers reported by our group together with representative references on the subject. We have also mentioned the article presenting clinical trial results from the EURIS phase 2B placebo control, carried out in 5 hospital centers from 1996-1998 on 240 patients (Gringeri A et al. Active anti-interferon- α immunization: A European-Israeli randomized, double-blind, placebo-controlled clinical trial in 242 HIV-1-infected patients (the EURIS Study). *J Acquir Immune Defic Syndr Hum Retrovirol* (1999) **20**:358-370), clinical trial preceded by a phase 1b. The EURIS trial showed a biological (CD4⁺ T-cell count) and clinical (AIDS related symptoms) benefits in vaccine responders. Phase 3 was not continued due to the arrival of triple therapy (cART) at the same time. Nevertheless, to satisfy the referee 1 demand we added one reference of Levy's group in the revised manuscript. This is the reference 28, which appears in the paragraph of the results section entitled "Elevated IFN α induces an abnormal frequency and phenotypic alterations of NK-cells and other innate cell types in non-EC but not in EC".

In the present study reported in the two companion manuscripts, we show that besides HIV direct effect per se, elevated IFN α is a key mediator for HIV pathogenicity. It is not just that IFN α is enhanced in AIDS; elevated IFN α triggers multiple pathogenic effects as we show here resumed in the companion paper

extended data Table 2. Moreover, elevated IFN α not only increases the expression of CCR5, the second HIV receptor on the surface of stimulated CD4⁺ T-cells but also inhibits the production of C-C chemokines as reported earlier by our group (Zagury D. et al.. Interferon alpha and Tat involvement in the immunosuppression of uninfected T cells and C-C chemokine decline in AIDS. Proc. Natl. Acad. Sci. U. S. A. 95, 3851–3856 (1998)). These chemokines protect cells from HIV infection (Cocchi F. et al.. Identification of RANTES, MIP-1 alpha, and MIP-1 beta as the major HIV-suppressive factors produced by CD8⁺ T cells. Science. 1995 Dec 15;270(5243):1811-5) by inducing internalization of the CCR5 receptor. All these direct and indirect actions of high IFN α promote the infection of CD4⁺ T-cells and the spread of HIV in other lymphoid foci. Beside HIV1 per se, elevated IFN α hampers activation of CD4⁺ T helper and follicular T helper cells which initiate the HIV specific immune reaction leading to production of neutralizing antibodies.

If the infection occurs in the presence of a high dose of virus, subsequently there will be a correlated elevated dose of IFN α , and an IFN α -induced increase of CCR5 expression on CD4⁺ T-cells and a progression of the pathogenesis leading to AIDS.

Minor points:

- Fig. 1D5: Please specify whether the DC are pDC or mDC.

Response : Figure 1B5 shows the frequency of dendritic cells (DC) which is defined by cytometry as a Lineage neg and HLA-DR + cell population (Lin- HLA-DR+). We have specified this in the revised manuscript on the graph and in the legend of the figure.

Within DC, pDC are characterized by the CD123⁺ CD11C⁻ phenotype and mDC by the CD123⁻ CD11C⁺ phenotype, as indicated in the extended data fig. 3 which presents the gating strategies of the different cell subpopulations studied.

- Fig. 2E: Please specify on how many donors the studies using increasing IFN-alpha concentrations were performed.

Response : The number of donors in the studies using increasing IFN α concentrations is indicated in the fig legends as follows: “Percentage of viable NK-cells (n=3) (E1) and frequency of CD56^{low} NK-cells (n=3) (E2) after 7 days of culture in presence of increasing doses of IFN α . Histograms showing the expression level of CD56 in CD56^{dim}/neg NK-cells (n=6) (E3), distribution of mature (grey) and terminal NK-cells (white) (n=6) (E4), expression levels of NKG2D in early NK-cells (n=3) (E5) and CD95 in mature NK-cells (n=3) (E6) after 3 days of culture in presence of IFN α .”

- Please explain abbreviations in the figures and table legends.

Response : We removed the abbreviations in the titles of the legends and indicated in the legend what the abbreviation corresponds to when it appears the first time.

We modify the text legends in the revised manuscript as follows :

- Legend Fig.1 : Title : non-EC, EC and HD were replaced by non-Elite Controllers, Elite Controllers and Healthy Donors.
- Legend Fig.2 : NCR, GrzB/perf and iKIR were replaced by Natural cytotoxicity receptors, Granzyme/Perforin and Inhibitory Killer Ig-Like Receptors.
- In the extended tables describing the patients we have added a legend to explain the abbreviations. Dx : Diagnosis, M : Male, F : Female, AA : Afro American, IDU : Injection Drug Use, HS : Homosexual, MSM : Men who have Sex with Men, NA : Non Available

-Methods: please provide reference numbers for ethical approval.

Response : The IRB number for the study in US is H-29331. In Liege (Belgium) the number of the agreement for the study is 2020/418. We added it in the material and method in the human samples section.

- P.15: Altfeld instead of Altflod

Response : Thank you for pointing out the typo. We fixed it in the text of the revised manuscript

- P. 18: was the “Composite cell phenotypic alteration score” used in Fig. 5D?

Response : The Composite cell phenotypic alteration score was used in the Fig. 3A4 and 4A5.

- Ref. 17: is the journal name correct?

Response : Thank you for pointing out the typo. We fixed it in the text of the revised manuscript as follows:

“Gringeri, A. et al. Active anti-interferon-alpha immunization: a European-Israeli, randomized, double-blind, placebo-controlled clinical trial in 242 HIV-1--infected patients (the EURIS study). J. Acquir. Immune Defic. Syndr. Hum. Retrovirology. 20, 358–370 (1999).”

Reviewer #2 (Remarks to the Author):

Buanec and colleagues propose that IFN alpha is a key determinant for HIV pathogenicity. The authors present numerous lines of evidence that IFN alpha undermines NK and other killer cell function. The results are provocative and, if correct, important to our understanding of HIV immunopathogenicity. My main concern lies with the nature of the cohorts. Do the authors have information of IFN alpha levels in the EC and non-EC subjects either at acute viremia or at set point? It would be interesting to know whether IFN levels at either of those episodes might predict subsequent disease sequelae. Furthermore, a discussion of treatment interruption is warranted and whether IFN levels at interruption reflect the rate of progression of the subjects prior to initiating therapy.

Response : We agree with referee 2 that it would be interesting to know whether IFN α levels at either acute viremia or at set point might predict subsequent disease sequelae.

We did not have the opportunity to follow IFN α level in Elite controller and non-controller patients from the onset of their infection. However, comparisons of SIV infection in primate species that develop AIDS-like disease and species without disease symptoms indicate that an elevated IFN α a serum concentration occur only during pathogenic infection in macaques, whereas natural SIV hosts, without disease progression, have weaker serum IFN α concentration levels (Mandl, J. N. et al. Divergent TLR7 and TLR9 signaling and type I interferon production distinguish pathogenic and nonpathogenic AIDS virus infections. *Nature Med.* 14, 1077–1087 (2008) ; Jacquelin, B. et al. Nonpathogenic SIV infection of African green monkeys induces a strong but rapidly controlled type I IFN response. *J. Clin. Invest.* 119, 3544–3555 (2009)). Similar findings have been made in individuals infected with HIV. Rapid progressors show stronger IFN α signatures than viraemic non-progressors (Rotger, M. et al. Comparative transcriptomics of extreme phenotypes of human HIV-1 infection and SIV infection in sooty mangabey and rhesus macaque. *J. Clin. Invest.* 121, 2391–2400 (2011)). These studies suggest a link between sustained serum IFN α levels and disease progression.

In 2017, Cheng et al published on the targeting of type I mediated activation in humanized mice models of HIV using IFNAR antagonists to restore immune functions and reduce HIV reservoirs. Cheng et al used an anti-IFNAR1 antibody in NRG-BLT humanized mice. cART was introduced 4 weeks after HIV1 challenge and anti-IFNAR1 was administered after seven weeks. cART was then interrupted after 12 weeks of administration. They demonstrated that blockade of the IFN receptor during the chronic phase of

infection reduces the levels of T cell activation, reduces expression of the inhibitory receptors PD-1 and TIM-3, and improves cytokine production by CD8⁺ T-cells. Above all, they show that type I IFN blockade during ART administration markedly reduced the frequency of cells harboring replication-competent HIV, the so-called “reservoir” and caused a delayed rebound of viremia after ART was discontinued (L. Cheng et al, Blocking type I interferon signaling enhances T cell recovery and reduces HIV1 reservoirs, *J. Clin. Invest.* 127 (2017) 269–279. doi:10.1172/JCI90745).